

# Modeling and evaluating the effects of irrigation on land-atmosphere interaction in South-West Europe with the regional climate model REMO2020-iMOVE using a newly developed parameterization

Christina Asmus[1], Peter Hoffmann[1], Joni-Pekka Pietikäinen[1], Jürgen Böhner[2], and Diana Rechid[1]

[1]Climate Service Center Germany (GERICS), Helmholtz Center Hereon, Hamburg, Germany
[2]Center for Earth System Research and Sustainability (CEN), Universität Hamburg, Hamburg, Germany

**Correspondence:** christina.asmus@hereon.de

**Abstract.** Irrigation is a crucial land use practice to adapt agriculture to unsuitable climate and soil conditions. Aiming for improving the growth of plants, irrigation modifies the soil condition, which causes atmospheric effects and feedbacks through land-atmosphere interaction. These effects can be quantified with numerical climate models, as has been done in various studies. It could be shown that irrigation effects, such as air temperature reduction and humidity increase are well understood and should not be neglected on local and regional scales. However, there is a lack of studies including the role of vegetation in the altered land-atmosphere interaction. With the increasing resolution of numerical climate models, these detailed processes have a chance to be better resolved and studied. This study aims for analyzing the effects of irrigation on land-atmosphere interaction, including the effects and feedbacks of vegetation. We developed a new parameterization for irrigation and implemented it into the REgional climate MOdel REMO2020, coupled with the interactive MOsaicbased VEgetation module iMOVE. Following this new approach of a separate irrigated fraction, the parameterization is suitable as a subgrid parameterization for high-resolution studies and resolves irrigation effects on land, atmosphere, and vegetation. Further, the parameterization is designed with three different water application schemes in order to analyze different parameterization approaches and their influence on the representation of irrigation effects. We apply the irrigation parameterization for South-West Europe including the Mediterranean region on 0.11° horizontal resolution for hot extremes. The simulation results are evaluated in terms of the consistency of physical processes. We found direct effects of irrigation, like a changed surface energy balance with increased latent and decreased sensible heat fluxes, and a surface temperature reduction of more than -4 K as mean during the growing season. Further, vegetation reacts to irrigation with direct effects, such as reduced water stress, but also with feedbacks, such as a delayed growing season caused by the reduction of the near-surface temperature. Furthermore, the results were compared to observational data showing a significant bias reduction in the 2 m mean temperature when using the irrigation parameterization.





## 1 Introduction

Land use and land use practices are anthropogenic forcings that were shown to influence regional climate. They can be defined as the modification of the land surface through anthropogenic changes of land cover types or land use practices that alter the land surface within one land cover type (Luyssaert et al., 2014). Through land-atmosphere interactions changes in the land conditions can affect the climate and cause feedback mechanisms, especially in the near-surface atmosphere levels (Jia et al.,

2019). Luyssaert et al. (2014) pointed out that under specific circumstances the effects of land use practices reach the same magnitude as land use and land cover changes effects, and therefore should not be neglected in climate studies.

We find different land use practices in agriculture such as tillage, fertilization, irrigation, etc. Irrigation is the land use practice that has the strongest impact on the climate (Kueppers et al., 2007; Lobell et al., 2009; Sacks et al., 2009). Further, irrigation is a common land use practice in agriculture to adapt to unsuitable climatic conditions. Using numerical models, irrigation

effects are studied on different scales, with different parameterizations, and for different regions. An overview can be found in Valmassoi and Keller (2022) who collected different irrigation modeling studies and identified the different aspects of irrigation parameterizations as sources for uncertainties of irrigation effects on the climate.

Global scale irrigation studies show different developments of irrigation effects in different regions in the world (Thiery et al., 2017, 2020; Sacks et al., 2009; Lobell et al., 2009; Puma and Cook, 2010; de Vrese and Hagemann, 2018). All studies found

near-surface and surface temperature reduction. Compared to observational data, using irrigation in the model Lobell et al. (2009) could eliminate the warm and dry bias of CLM. De Vrese and Hagemann (2018) showed that irrigation has remote effects in more than 100 km distance of the irrigated area. Further, multiple studies showed that irrigation effects are more pronounced on local and regional scales (Sacks et al., 2009; Kueppers et al., 2007; Valmassoi et al., 2019). In particular, high-resolution studies on regional scale require an accurate representation of the land surface and soil processes to represent local

and regional climatic patterns (Hagemann et al., 1999). For example, Saeed et al. (2009) showed the irrigation effects on the summer monsoon in India, which is weaker due to a smaller land-sea-temperature gradient. Also, Tuinenburg et al. (2014) studied irrigation effects in India and found a shift in the precipitation pattern through the additional moisture in the atmosphere. Valmassoi et al. (2020a) studied irrigation effects in the Po Valley on convection-permitting scale and found an increase of precipitation at irrigated areas. As Thiery et al. (2020) on global scale, Kueppers et al. (2007) pointed out the potential of

irrigation to mask the warming effects of greenhouse gases on regional scale for a study in California. Further, Kueppers et al. (2007) showed that irrigation effects follow a seasonality. During the growing season, the effects are most pronounced, and for dry periods the effects are stronger than for wet periods. Thiery et al. (2020) and Jia et al. (2019) point out that the near-surface temperature reduction through irrigation decreases the probability of hot extremes. With these characteristics, irrigation becomes a potential adaptation measure to climate extremes, not only to water stress that plants experience during droughts but

in addition, it can be implemented to reduce the intensity of heat waves.

The simulated effects of irrigation on the land-atmosphere interaction depend on one hand on the amount of irrigation, as pointed out by Valmassoi et al. (2019), and on the other hand on the design of the parameterization itself. The irrigation amount is driven by the soil hydrology of the model. Multiple models represent the soil hydrology using a layered scheme and





prescribe observed irrigation amounts (Valmassoi et al., 2019; Puma and Cook, 2010; Ozdogan et al., 2010; Yao et al., 2022).

For models using a bucket scheme (Boucher et al., 2004; de Vrese and Hagemann, 2018), observed irrigation values might not fit due to the deep bucket. Therefore, the irrigation parameterizations are designed with thresholds based on specific model-internal, physical values e.g. values of the maximum water-holding capacity of soil, field capacity, LAI, or photosynthesis rates, to determine the irrigation amount, or the irrigation start and end. However, using such a model-internal, physical threshold rather than a prescribed irrigation amount often leads to an overestimation of the effects (Kueppers et al., 2007). Therefore,

Thiery et al. (2017) added a water limit for the available irrigation amount to reach realistic values, and Leng et al. (2017) added a water source and closed the hydrological cycle. For representing irrigation in a climate model, it is recommended to have a separate soil column for irrigation (Lobell et al., 2009; Ozdogan et al., 2010; Thiery et al., 2017) and represent irrigated areas on subgrid-scale. Another aspect of representing irrigation in a climate model is the irrigation method. Irrigation methods differ in their water application. Mostly, irrigation is represented as an increase of soil moisture, neglecting canopy interactions

(Sacks et al., 2009; Lobell et al., 2009; Ozdogan et al., 2010; Thiery et al., 2017; de Vrese and Hagemann, 2018). Newer studies consider canopy effects which are caused by e.g. sprinkler irrigation (Valmassoi et al., 2019; Leng et al., 2015; Yao et al., 2022). However, on regional scale, the differences in the irrigation effects between different irrigation methods remain small and can be neglected (Valmassoi et al., 2019).

For most methods, irrigation affects the land surface, altering the exchange processes through land-atmosphere interaction. On

high resolution, a more detailed representation of the land surface and its processes is possible. An important driver of these land processes is vegetation which is, such as the soil, the land surface, and the atmosphere, also affected by irrigation. However, there is a lack of high-resolution climate studies which include the irrigation effects on vegetation and its feedback on the atmosphere, soil, and surface. This study aims to represent irrigation effects in the model system REMO2020-iMOVE which represents land, atmosphere, and vegetation processes interactively. Whereas Saeed et al. (2009) analyzed large-scale irrigation

effects with REMO2009, this study aims for a detailed representation of irrigation aspects and conducts high-resolution experiments. Thus, we implement a new irrigated fraction and represent irrigation on subgrid-scale. Our model region is South-West Europe with a focus on one of the most intensely irrigated areas in Europe, the Po Valley. After we described the model and the data that we used for this study in section 2, we introduce our new irrigation parameterization (section 3). In section 4 we apply the new irrigation parameterization and evaluate it with the consistency of physical processes as well as with the comparison of

observational data. We point out some limitations of our parameterization in section 5 and give concluding remarks in section 6.

## 2 Model and data

### 2.1 The model REMO2020-iMOVE

For this study, the regional climate model REMO2020 was used (Pietikäinen et al., in prep.). REMO is developed as a hydro-

static atmospheric circulation model based on the primitive equations of atmospheric motion at the Max-Planck-Institute for Meteorology, Hamburg, Germany (Jacob, 1997, 2001). It combines parts of the Europa Model (EM) of the German Weather



Service (Majewski, 1991) and the physical parameterizations of ECHAM4 (Roeckner et al., 1996). With time, REMO was further developed and received additional features such as dynamic vegetation cover (Rechid and Jacob, 2006), glaciers (Kotlarski, 2007), lakes (Pietikäinen et al., 2018), a non-hydrostatic extension to the hydrostatic core (Goettel, 2009), and an interactive
mosaic-based vegetation module (iMOVE) (Wilhelm et al., 2014). For this study, in particular, the land surface parameterizations are of interest.

The surface of one model grid box in REMO2020 is represented with the tile approach in which the subgrid fractions land, water, representing sea and lakes, and sea ice are introduced (Semmler, 2002). Using the lake module FLake (Pietikäinen et al., 2018), a separate lake subgrid fraction is added. In total, the fractions sum up to 100 % of the surface of a model grid box.
Whereas the land fraction is constant, the sea-ice fraction can vary, thereby changing the water fraction. For each fraction turbulent surface fluxes and radiation fluxes are calculated and averaged at the lowest atmospheric level using weighted means with respect to the fraction area of the model grid cell. Using the bulk transfer relations with transfer coefficients from the Monin-Obukhov similarity theory with a higher order closure scheme, the turbulent fluxes of momentum and heat are calculated (Kotlarski, 2007). The exchange processes between the atmosphere and surface are determined by the vegetation's coverage.
Since the vegetation's physiology depends strongly on seasonal cycles, the variations are included for the vegetation fraction, the LAI, and the background albedo (Rechid and Jacob, 2006). To improve the vegetation's representation and its effects on the atmosphere, the iMOVE modules of REMO2009-iMOVE (Wilhelm et al., 2014) are implemented into REMO2020. Multiple elements of iMOVE are based on the dynamic land surface scheme JSBACH (Raddatz et al., 2007; Wilhelm et al., 2014). It represents the land cover with tiles of plant functional types (PFTs) using the Holdridge ecosystem classification scheme
(Wilhelm et al., 2014). For this experiment, the definition and distribution of PFTs are based on the land cover maps of the European Space Agency Climate Change Initiative (ESA-CCI) (Reinhart et al., 2022; Hoffmann et al., 2022). The PFTs interact dynamically with the atmosphere and the soil, leading to varying phenology. Here, soil moisture and air temperature are important driving factors (Wilhelm et al., 2014). Furthermore, in REMO2020-iMOVE soil moisture determines the soil albedo following the findings of Peterson et al. (1979) and model adjustments of Wilhelm et al. (2014). As a result, the soil albedo is
represented with a negative exponential relationship with the soil moisture.

The heat budget of the soil is represented with a 5-layer scheme. The heat transfer is calculated with diffusion equations for five discrete layers. For solving the equations, it is assumed that the heat flux is zero at the lowest boundary. The heat transfer between the layers is mainly driven by the heat conductivity and heat capacity of the soil type, which vary with soil moisture. The soil hydrology consists of three water storage reservoirs: soil, skin reservoir (vegetation), and snow, for which budget
equations are solved. The reservoirs are altered by precipitation, interception, dew, evapotranspiration, snow melt, runoff, infiltration, and drainage (Kotlarski, 2007). Precipitation is split by the improved Arno-Scheme (Dümenil and Todini, 1992) into surface runoff and infiltration considering subgrid-scale heterogeneous field capacities of the land surface within one grid cell (Hagemann, 2002). The field capacity in REMO is at the level of the maximum water-holding capacity (wsmx) which is based on the global dataset of land surface parameters (LSP) by Hagemann et al. (1999). Once the soil moisture reaches wsmx,
runoff occurs. Infiltration fills up the soil moisture reservoir which is represented as a simple bucket scheme with subsurface drainage. The drainage is led by the ratio of the soil moisture and wsmx. Drainage occurs for soil moisture larger than 5 % of



wsmx. Between 5 % and 95 % of wsmx, drainage is slow. If the soil moisture is larger than 90 % of wsmx, the drainage is fast (Kotlarski, 2007).

Water can leave the soil moisture reservoir through evapotranspiration depending on vegetation characteristics and atmospheric conditions. For bare soil, evaporation takes place from the upper 10 cm. Subsurface water leaves the soil moisture reservoir only through transpiration by vegetation or drainage. At the surface or soil, there are no lateral flows of water within REMO2020 (Wilhelm et al., 2014).

## 2.2 Irrigation dataset

For an estimation of the spatial distribution of irrigated areas, the Global Map of Irrigated Areas Version 5 (GMIA5) by Siebert et al. (2013a) is used. The GMIA5 describes the area equipped for irrigation as well as the area actually irrigated on a resolution of 5 arcmin (0.083333 decimal degrees). It was developed at the Johann Wolfgang Goethe University, Frankfurt (Main), Germany, by Doell and Siebert (1999). Through cooperation with the Rheinische Friedrich-Wilhelms-Universität, Bonn, Germany, and the Land and Water Division of the FAO, GMIA is constantly improved and updated. The dataset is mainly based on AQUASTAT, the FAO's information system on water and agriculture. The data is collected from national and subnational water resources and irrigation plans, statistics, yearbooks, and FAO technical reports. This information is combined with geospatial information on the position and extent of the irrigated area. The statistical data refers to the years from 2000 to 2008, with the reference year depending on the country. The quality of GMIA5 was assessed by the density of used subnational irrigation statistics and by the density of the available geospatial records on the position and extent of irrigated areas (Siebert et al., 2013b).

For our study, we chose the data of the "area equipped for irrigation" of the GMIA5 due to its better data quality (Siebert et al., 2013b) as well as due to our study's purpose of showing maximal possible irrigation effects.

## 2.3 Observation data for evaluation

The Italian Institute for Environmental Protection and Research (ISPRA) established a database for meteorological observation data for Italy named SCIA (*ital.* Sistema nazionale per la raccolta, l'elaborazione e la diffusione di dati Climatologici di Interesse Ambiental). SCIA works as a framework of the national environmental information system and combines data from national and regional networks, agro-meteorological stations (UCEA-RAN), as well as from hydro-meteorological stations, and from tide gauge networks. The data is updated once per year and undergoes a quality check. Climate indicators are available for different time scales such as means of 10 days, months, or years (Desiato et al., 2011) and are freely available at the SCIA website (www.scia.isprambiente.it). For this study, the monthly means of the daily mean, maximum, and minimum 2 m temperature for the year 2017 are used.



## 3 Development of the irrigation parameterization

### 3.1 Implementation of a new irrigated land subfraction and a new PFT into REMO2020-iMOVE

In REMO2020-iMOVE, soil processes are defined for land fractions (Kotlarski, 2007). Irrigation influences soil and surface
directly and is a new local process to implement into REMO2020-iMOVE. Since it affects the land fraction, we implement a
new irrigated land fraction based on the "area equipped for irrigation" from the GMIA5 (subsection 2.2). Before using it in
REMO-iMOVE, GMIA5 has to be adapted to the desired resolution and geographic projection. The new irrigated land fraction
in REMO2020-iMOVE is a new land fraction that can be understood as a subfraction of the land fraction (Fig. 1). All soil,
surface, and vegetation processes are calculated for both land fractions, except for irrigation, which is applied exclusively to
the irrigated land fraction.

As land cover, we implement a new PFT named "irrigated cropland" on the irrigated land fraction. The properties of "irrigated
cropland" are based on the properties of the "cropland" PFT of the non-irrigated land fraction. REMO2020-iMOVE is able to
distinguish between the photosynthesis path of cropland PFTs (C3 or C4), however, it does not distinguish between different
crop types. In our case "irrigated cropland" is the only irrigated PFT, and therefore, the only PFT on the irrigated land fraction.
With the separation into an irrigated and non-irrigated land fraction and the new PFT, we ensure that the irrigation process
is only applied to areas that are truly irrigated. Having a separate irrigated land fraction gives a detailed representation of the
heterogeneity of the surface and the irrigated areas, which is an advantage for high-resolution and small-scale irrigation studies
such as on the European continent where irrigated areas are rather scattered.

The implementation of the new irrigated land fraction is done during the model initialization. The irrigation process addresses
exclusively the irrigated land fraction and starts after the soil hydrology. After the irrigation, the vegetation processes start,
which are partly driven by the moisture content in the soil and in the atmosphere (Fig. A1).

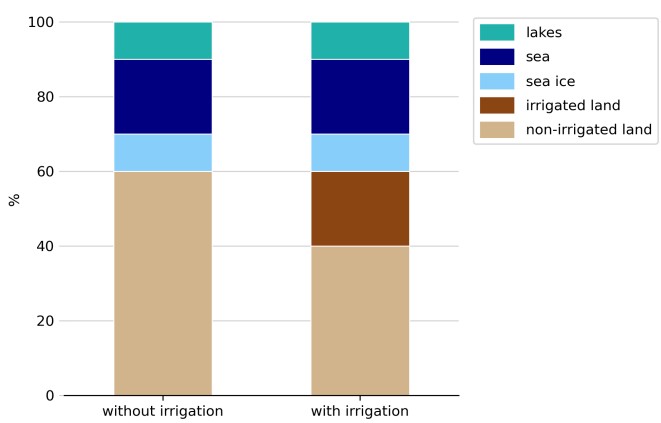

**Figure 1.** Fractions of one example model grid cell in REMO2020-iMOVE + FLake with irrigation.





## 3.2 Irrigation module and its different water application schemes

We implemented the new irrigation module into REMO2020-iMOVE, which can be turned on and off. The irrigation module determines where, when, and how irrigation will be applied. Irrigation is applied exclusively on the irrigated fraction (subsec-
tion 3.1), which defines the area equipped for irrigation from Doell and Siebert (1999)(subsection 2.2). Using an adjustable threshold (irrthr) on the soil moisture, the irrigation module determines the grid cells with irrigation requirements and creates a daily irrigation mask. This determination is carried out during the growing season because only then plants require irrigation. The growing season depends on the location and a growing degree threshold (Wilhelm et al., 2014). With fulfilling the require-ments for irrigation (Fig. 2), the water application starts. For parameterizing channel irrigation, the water is added directly to the
soil and increases the soil moisture. Here, we assume an infinite water supply. The water application and the irrigation amount strongly depend on the soil hydrology parameterization of the climate model, as well as on the ambition to be close to reality. Therefore, we implemented three different water application schemes, which can be used for different purposes (Table 1).

The "prescribed irrigation" scheme applies a prescribed amount of water within a prescribed time. The prescribed water amount will be equally distributed over each time step during the irrigation time. The water amount can be based on observed
irrigation values, but also extreme situations, such as a limited water supply or a huge water supply can be simulated. However, having a simple soil hydrology parameterization, such as the bucket scheme in REMO2020-iMOVE, suitable values for the prescribed water amount might differ from observed irrigation amount values leading to a necessary adjustment to reach realistic soil moisture conditions in the model. The prescribed water amount is a universal value, which will be added to the irrigated fraction in all model grid cells, which fulfill the irrigation requirements (Fig. 2). Further, the water amount in the
model does not depend on the crop type, since REMO-iMOVE does not distinguish between different crop types.

The "flexible time irrigation (flextime)" is based on a prescribed soil moisture target and open irrigation time. For each grid cell, the water amount is calculated that is necessary to reach the soil moisture target. Again, the water amount is equally distributed over each time step within the prescribed time. Once the soil moisture target is reached, the water application stops regardless of the irrigation time. For this approach, a soil moisture target has to be chosen in relation to wsmx of the soil.

The "adaptive irrigation" is also based on a prescribed soil moisture target and a limited irrigation time. Again, for each grid cell, the water amount is calculated that is necessary to reach the soil moisture target. The water amount added every time step follows a relaxation approach and considers the number of irrigation time steps remaining (Eq. 1).

$$ws_{t+1} = ws_t + \frac{irrtar * wsmx - ws_t}{nirr_t} \tag{1}$$

where:

$ws$ = soil moisture

$irrtar$ = irrigation target

$wsmx$ = maximum water-holding capacity

$nirr$ = number of remaining irrigation time steps

$t$ = time step



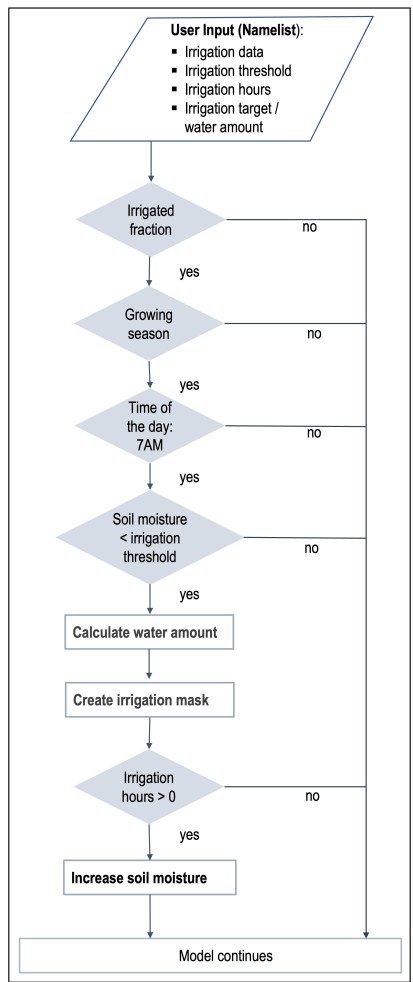

**Figure 2.** Irrigation process flow in REMO2020-iMOVE.

## 4   Results and evaluation of the parameterization

### 4.1   Experiment setup

We employ REMO2020-iMOVE for our model domain covering South-West (SW) Europe and the Mediterranean region, including some of the most intense irrigated areas such as the Po Valley or the Ebro Basin (Fig. 3). In 2017, SW Europe experienced exceptionally high temperatures, starting in June and reaching a heatwave in early August (subsection 4.3). This is the period we chose for our simulation because, first, irrigation is most important for agriculture during hot periods and, second, the effects of irrigation are most pronounced (Kueppers et al., 2007).

We conduct three one-month simulations to test the different water application schemes (T1 - T3, Table 2) for June 2017. Based on these short tests, we decide on one water application scheme, to conduct a one-year simulation (S1) to analyze the





**Table 1.** Properties of the different water application schemes in REMO2020-iMOVE.

|  | Prescribed irrigation | Flexible time irrigation (flextime) | Adaptive irrigation |
|---|---|---|---|
| Namelist variables | – irrigation time<br>– irrigation amount | – approx. irrigation time<br>– soil moisture target | – irrigation time<br>– soil moisture target |
| Irrigation amount ($irrw$) | prescribed in namelist | $irrw = \Delta ws$<br>$= irrtar * wsmx - ws_t$ | $irrw = \Delta ws_t$<br>$= irrtar * wsmx - ws_t$ |
| Irrigation stop | limited to irrigation amount | reaching soil moisture target | reaching soil moisture target after irrigation time |
| Water application | evenly distributed over each timestep during irrigation time | evenly distributed over each timestep during irrigation time | adaptive distributed over each time step during irrigation time |

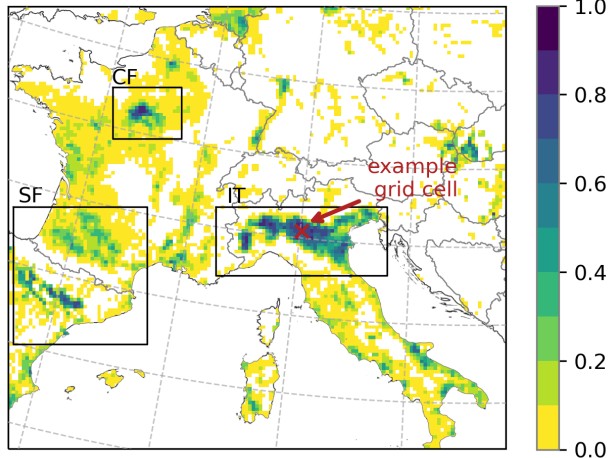

**Figure 3.** Model domain (129x145) grid cells with the fraction of irrigated areas and the analysis regions in Italy (IT), in northern Spain and southern France (SF), and in Central France (CF). The example grid cell of subsection 3.2 is pointed out.

effects of irrigation in the course of the year 2017. Simulation S0 is our baseline experiment and does not apply irrigation. All our simulations use a rotated grid with the rotated north pole at 39.25 N, 162 W, and have a horizontal resolution of 0.11°. We use ERA5 on 50 vertical levels as boundary data and set the time step to 60 s. S0 and S1 are started from 01/01/2017 with the soil variables in equilibrium state from a previous REMO simulation. The test simulations are started as a restart from our baseline experiment S0 from 01/06/2017. Table 2 summarizes the settings for the different test simulations T1 to T3, as well

as for the one-year simulations S0 and S1.

     T1, T2, and T3 test the water application schemes prescribed, flexible time (flextime), and adaptive to estimate their effect




on the development of irrigation effects. For all three test simulations, the irrigation threshold for the soil moisture is set to 0.75 of wsmx. For the model, this threshold is important because, from 0.75 of wsmx, the vegetation processes have optimal conditions to develop.

T1 uses a prescribed irrigation amount of 150 mm day$^{-1}$ which is evenly distributed over the irrigation time in all grid cells with irrigation requirements. We selected 150 mm day$^{-1}$ as irrigation amount from experience using the bucket scheme as soil hydrology (subsection 3.2). Following Bjorneberg (2013) channel irrigation is performed for 6 to 12 hours, we chose 10 h irrigation time for our experiment. T2 is testing the water application scheme with flexible time. This water application scheme is driven by the difference between the soil moisture at irrigation start at 7 AM LT and the irrigation target. We set the irrigation

target to the maximum, the maximum water-holding capacity. T3 is testing the adaptive water application scheme. As in T2, the irrigation target is set to the maximum. The irrigation time is set to 10 h as in T1. Since the test simulations are started as restarts from S0, the irrigation module detects grid cells with irrigation requirements from 01/06/2017.

**Table 2.** Simulation setup for the different water application schemes tests and the one-year simulation.

| simulation | simulation period (in 2017) | boundary data | initial condition (in 2017) | water application scheme | irrthr (as fraction of wsmx) [-] | irrtar (as fraction of wsmx) [-] | irrigation duration [h] | preset irrigation water [mm] |
|---|---|---|---|---|---|---|---|---|
| T1 | 01/06-30/06 | ERA5 | restart from S0 | prescribed | 0.75 | - | 10 | 150 |
| T2 | 01/06-30/06 | ERA5 | restart from S0 | flextime | 0.75 | 1.0 | - | - |
| T3 | 01/06-30/06 | ERA5 | restart from S0 | adaptive | 0.75 | 1.0 | 10 | - |
| S0 | 01/01-30/12 | ERA5 | ERA5[*] | not irrigated | - | - | - | - |
| S1 | 01/01-30/12 | ERA5 | ERA5 [*] | adaptive | 1.0 | 1.0 | 10 | - |

[*]with soil conditions in equilibrium state from previous REMO simulation

After testing the irrigation parameterization with its different water application schemes, our experiment aims to investigate irrigation effects on multiple variables and processes in the model system REMO2020-iMOVE and to check their physical

consistency over the course of one year. We quantify the irrigation effect by the difference between one simulation with the irrigation parameterization turned on and our baseline simulation with the irrigation parameterization turned off (S0). In S1 the irrigation process starts with the growing season of crops in the model domain. It only turns off, once the crops are harvested. In the course of the year, we analyze delayed irrigation effects and how they affect hot extremes. S1 applies the adaptive water application scheme with the irrigation threshold and the irrigation target at wsmx, leading to the maximum irrigation effects.



## 4.2 Testing the different water application schemes

Figure 4 shows the irrigation process with the different water application schemes for one representative, irrigated grid cell in the Po Valley (63,85) (Fig. 3) for the first irrigation day, 01/06/2017. We use a single grid cell to analyze the development of soil moisture in detail without any averaging. The soil moisture is at 0.47 of wsmx, leading to irrigation from 7 AM LT. For the prescribed water application scheme (T1) the soil moisture increases linearly until the irrigation time is finished, in this case at 5 PM LT (Fig. 4a). During the irrigation time, the same water amount is added for every time step. In the example grid cell, it is 15 mm h$^{-1}$ (Fig. 4b). At the end of irrigation, soil moisture reaches 0.87 of wsmx and stays close to this level until the end of the day (Fig. 4a).

For the simulation using the flextime water application scheme (T2) the soil moisture increases linearly until the irrigation target is reached after 301 minutes (Fig. 4a). As in T1, the same water amount is added to the soil moisture for each time step. However, the amount of added water is driven by the difference between the soil moisture at 7 AM LT and the irrigation target, leading to a higher added water amount per time step than in T1 (40 mm h$^{-1}$, Fig. 4b).

The adaptive water application scheme causes a non-linear increase of the soil moisture converging to the irrigation target and reaching it in the last time step of the irrigation time (Fig. 4a) which is set to 10 h. The water application adjusts itself in each time step depending on the difference between the actual soil moisture and the irrigation target and on the remaining time steps with irrigation (Eq. 1). Thus, for the first irrigation time steps, when the difference is the greatest, the water amount added is the greatest with 38 mm h$^{-1}$. It decreases with the following irrigation time steps (Fig. 4b).

Comparing the irrigation amount used in June (Fig. 5), the water amount added in T2 and T3 is very similar (max. 380 mm month$^{-1}$), which is also shown in the distribution of the irrigation water amount in Fig. 5d. The irrigation water amount added by the prescribed scheme in T1 is in particular in grid cells in the Po Valley, the Ebro Basin, and Southern Italy larger than in T2 and T3. The prescribed scheme also reaches the highest irrigation water value (max. 450 mm month$^{-1}$, Fig. 5d). The reason for these differences is that the prescribed water application scheme stops the irrigation in one day once the prescribed irrigation amount is finished within the prescribed irrigation time, regardless of the saturation of soil moisture. This leads to multiple irrigation requirements in June, once the soil moisture drops below the irrigation threshold, turning on irrigation. Using the flexible time (T2) and the adaptive water application scheme (T3), in most grid cells only one irrigation event is necessary in June, whereas using the prescribed irrigation scheme (T1) required up to three irrigation events always adding the same prescribed irrigation amount (Fig. A2).

The overall effects of the three water application schemes as monthly mean values are similar (Fig. A3, Fig. A4). Therefore, we select only one scheme to analyze further the effects of irrigation on the regional climate. The water application scheme selected is the adaptive water application scheme, since it has multiple advantages. First, it reaches smoothly the irrigation target and takes into account the actual soil moisture of each grid cell and the remaining irrigation time steps. Second, the relaxation method is a common method in climate modeling. And third, the adaptive water application scheme is the user-friendliest scheme of the three schemes because it does not require experience values of the irrigation amount depending on



the soil hydrology of the climate model.

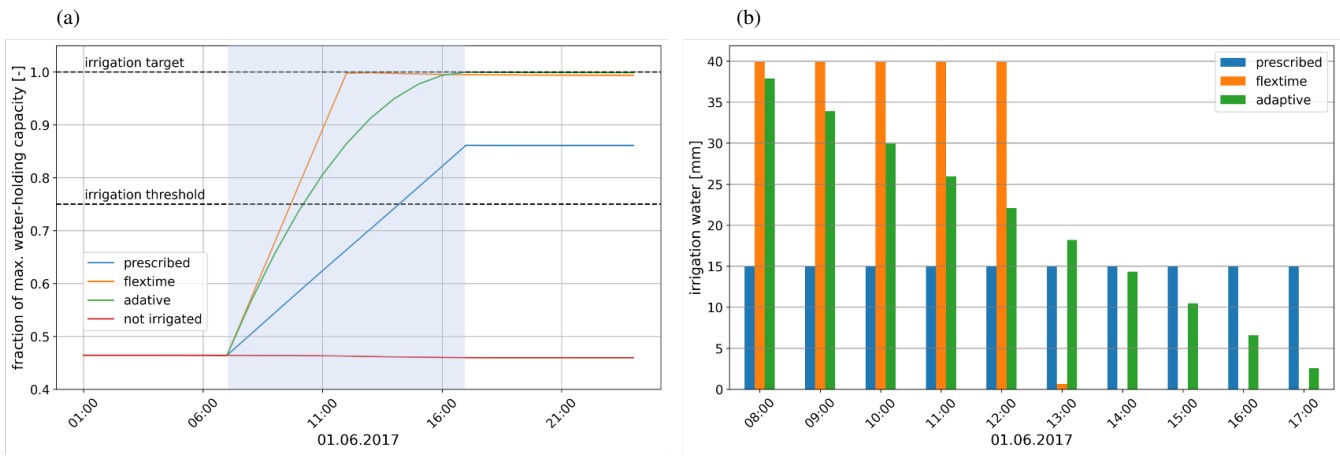

**Figure 4.** Irrigation process on the first irrigation day (01/06/2017) using the different water application schemes in one representative example selected grid cell (63,85). Settings: irrigation threshold at 0.75 of wsmx, irrigation target at wsmx, irrigation time for 10 h. Blue shaded is irrigation time. a) soil moisture as fraction of wsmx and b) irrigation water used.

**4.3  Simulated meteorological conditions during spring and summer 2017 with REMO2020-iMOVE**

SW Europe and the Mediterranean region experienced dry and warm weather during spring and summer in 2017. According to EOBS data, spring (MAM) was 1.7 °C warmer than the reference period 1981-2010. During summer (JJA) several heat waves occurred in SW Europe, as well as in the Balkans (Copernicus Climate Change Service, 2022). One of the first heat waves hit SW Europe in June (Sánchez-Benítez et al., 2018), in particular Spain and France. Another heat wave developed at the
beginning of August 2017 in southern Europe, this time in particular in Spain, France, Italy, and the Balkans (Kew et al., 2019) causing several wildfires (Copernicus Climate Change Service, 2022).

The warm and dry meteorological conditions in spring as well as the hot conditions in summer are represented in the REMO2020-iMOVE simulation in the southern part of the model domain (Fig. 6, Fig. 7). Within spring and summer, the months of April, May, and June (AMJ) are of particular interest because irrigation is linked to the growing season and these
months will be fully irrigated. Therefore, we analyze the meteorological conditions during AMJ (Fig. 6a-c) as well as for the heat wave in August (Fig. 6d-f) to investigate delayed irrigation effects in the model without active irrigation. The mean 2 m temperature distribution for AMJ follows a North-South pattern as well as the topography. The highest values of up to 25 °C occur in the river valleys of the Po, the Ebro, and the Garonne and Adour (Fig. 6a). In these valleys, the soil moisture is the lowest in the model domain (Fig. 6b), most precipitation, which could fill up the soil moisture, falls in the mountains of the
Alps, Pyreenes, Central Massif, and Dinaric Alps (Fig. 6c). Fig. 6d-f show the simulated mean conditions during the heat wave from 03/08/2017 - 05/08/2017. The highest temperatures of up to 40 °C are reached in Italy as well as in the Balkans (Fig. 6d).



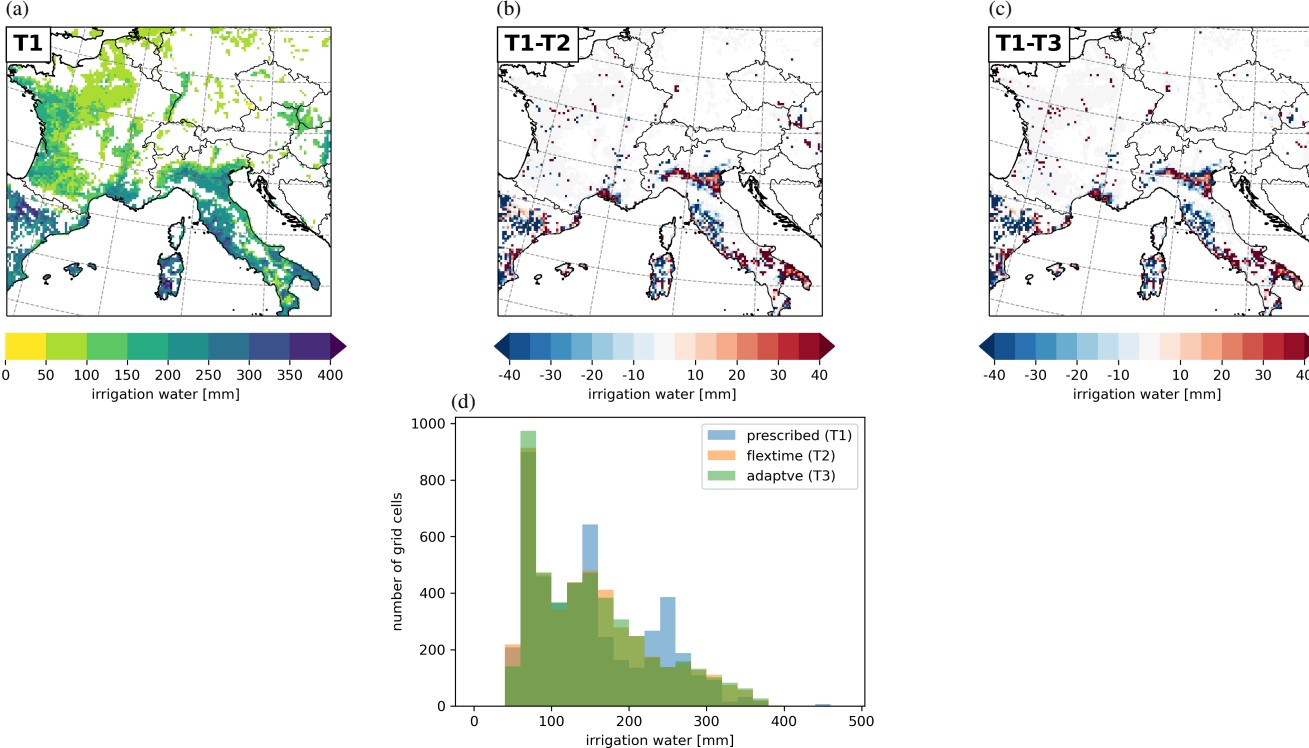

**Figure 5.** Irrigation water used for the different water application schemes in June 2017, a) prescribed (T1), b) as difference between prescribed (T1) and flextime (T2), c) as difference between prescribed (T1) and adaptive scheme (T3), and d) distribution of irrigation water in irrigated grid cells.

In the northern part of the model domain, the heat wave was not present. Fig. 7 shows the evolution of the meteorological conditions from April until August in the three analysis regions (Fig. 3). Within the course of the year and the beginning of summer (June), the soil moisture drops in all three analysis regions. Since the soil properties differ, the soil moisture differs in 290 the analysis regions with higher values in CF and the lowest values in IT. Due to low precipitation rates, in particular in IT and SF, the soil moisture cannot be filled up in the analysis regions. The evolution of temperature in the analysis regions shows hot summer periods (Fig. 7d-f). Whereas IT experienced the most extreme heat wave at the beginning of August, CF experienced its highest temperatures at the end of August. The heat wave in IT lasted for 3 days in accordance with EOBS data (Copernicus Climate Change Service, 2022). As a region mean the daily 2 m maximum temperature reaches up to 35 °C and the daily 2 m 295 minimum temperature up to 25 °C.




**Figure 6.** Simulated mean meteorological condition with REMO2020-iMOVE for a) 2 m temperature during AMJ, b) soil moisture during AMJ, c) monthly mean of summed precipitation for AMJ, d) 2 m temperature during heat wave (03/08/2017-05/08/2017), g) mean soil moisture during heat wave (03/08/2017 - 05/08/2017), f) mean of summed precipitation during heat wave (03/08/2017 - 05/08/2017).

## 4.4 Process analysis of irrigation effects

To understand the effects of the irrigation parameterization, we analyze the results of the extreme scenario, in which the irrigation threshold and the irrigation target are wsmx of the soil, this is the maximal possible value of irrigation effects. This setting causes everyday irrigation during the growing season resulting in soil moisture close to wsmx in the irrigated grid cells.

### 4.4.1 Effects on soil and surface fluxes

The irrigation effects are analyzed in their spatial distribution as well as in their occurrences in the diurnal and annual cycles. The soil moisture is directly increased by the parameterization, which is shown in Fig. 8a as a mean of the irrigated months April, May, and June (AMJ). Depending on the local wsmx of the soil and the actual soil moisture, the irrigation requirement



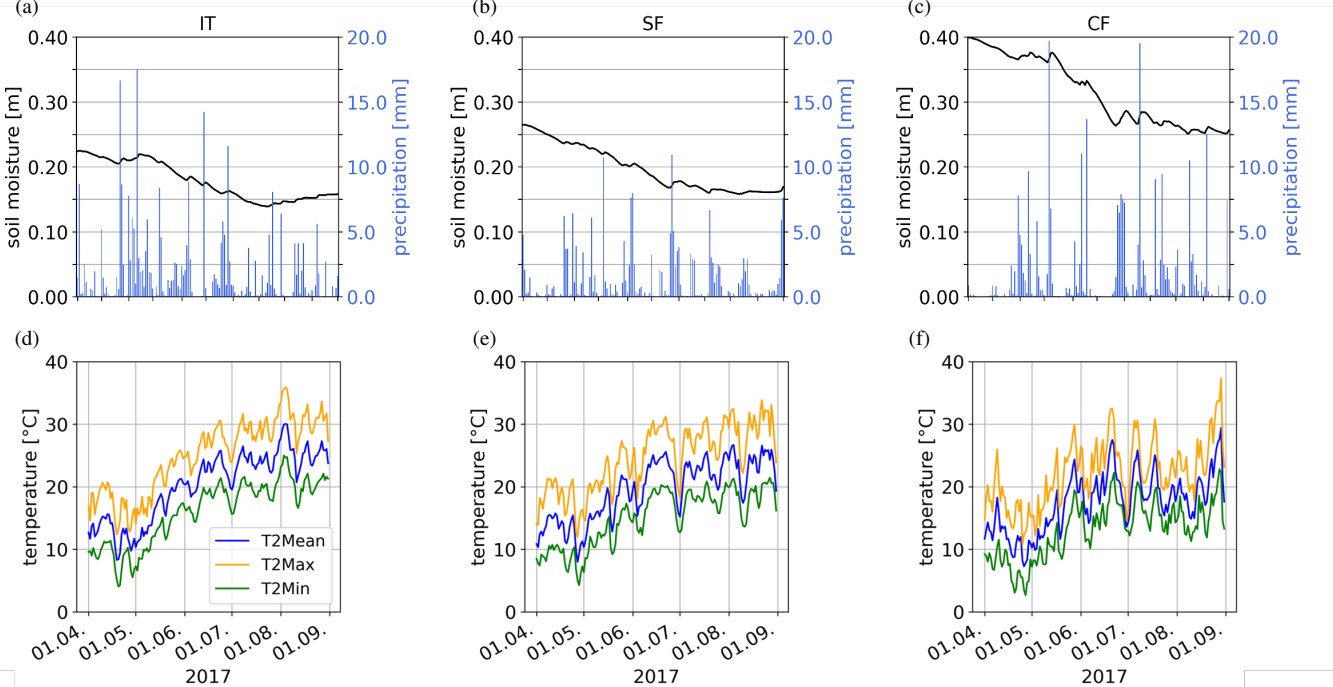

**Figure 7.** Meteorological conditions as spatial means of the analysis regions IT, SF, and CF in 2017 for AMJJA for a-c) soil moisture and precipitation, d-f) 2 m temperatures.

in each grid cell differs from each other. Fig. 8a shows a North-South gradient of the irrigation requirement with the highest

values of up to 600 mm in the South like in the Ebro basin in Spain and the Balearic Islands as well as in Italy in Sardinia, Puglia, Lazio and the Po Valley. In the northern irrigated areas such as in France, the irrigation requirement is on average 200 mm for AMJ in the model.

Irrigation effects appear in the diurnal cycle of the soil moisture (Fig. 8b). The irrigation start time is at 7 AM LT which increases the soil moisture, slowly at first, then faster as we get closer to the end of the irrigation end time. At 5 PM LT, the

maximum irrigation effect is reached for soil moisture with an increase of 202 mm as spatial average of irrigated areas in the model domain during AMJ.

In the annual cycle, the irrigation effects start to occur from March and increase until July (Fig. 8c). In July, the irrigation effects of the soil moisture reach +300 mm as a monthly average of all irrigated areas in the model domain. In most areas of the model domain, the growing season stops in July. Therefore, the irrigation effects decrease from August until the end of the

year. Nevertheless, the soil moisture remains at a higher level than in the simulation without irrigation due to irrigation in the months before.

The effects of irrigation occur in different layers of the soil temperature as well as in the surface temperature (Fig. 8d-f). In general, irrigation reduces the surface temperature (Fig. 8d). The spatial distribution of that cooling follows the changes in



the surface fluxes (Fig. 9). The strongest cooling effect occurs in the Ebro Basin and in the southern Po Valley with -4 K as
a mean value in AMJ. The cooling at the surface propagates to the deeper layers of the soil, which is shown in the diurnal
and annual cycle of the soil temperatures at different depths (Fig. 8e-f). The upper three layers up to a depth of 1.232 m are
influenced by the surface processes. In Fig. 8e, the effects on the upper soil temperature from 0.0 m to 0.065 m follow the solar
radiation, reaching its maximum cooling by irrigation at 1 PM LT with -3.2 K. The temperature of the second soil layer reacts
time-shifted and reaches its maximum cooling by irrigation at 6 PM LT with -1.9 K. The levels from 0.319 m depth do not
show a diurnal cycle anymore, however, they show a cooling between -0.05 K and -1.4 K.

Since soil reacts inertly, the irrigation effects on the soil temperature throughout the year 2017 are analyzed with monthly
mean values (Fig. 8f). The same order of the magnitude of the cooling effect is shown for the different temperature layers as it
is in the diurnal cycle (Fig. 8e). The upper four layers react immediately to irrigation and show a cooling from March where
the upper layer at the surface reaches a cooling of up to -1.5 K and the fourth layer at 1.232 m depth reaches a cooling of -0.05
K. The two upper layers reach their maximum cooling effect in April, whereas the third layer reaches its maximum cooling
effect in July, the fourth layer in August, and the fifth layer in December. This time shift shows the inertial reaction of the soil
temperature. The cooling of the three upper soil layers develops in spring (from March) and summer months until the harvest
in July begins in wide areas of the model domain. From August the cooling effect is reduced in the upper three layers.

In general, the cooling in the soil temperature is explained by mainly two processes. First, surface processes like the enhanced
latent heat flux and evaporation cool the surface temperature. This cooling propagates slowly in deeper levels. Secondly, the
cooling is caused by the soil moisture-dependent heat capacity and thermal conductivity, which increase with higher soil
moisture (Eggert, 2011). This leads to faster signal transmissions and thus to faster cooling rates.

In the irrigated months AMJ, irrigation leads to an increase of evapotranspiration with the maximum in the Ebro Basin, in
Sardinia, and in Lazio with an evapotranspiration increase of up to +150 mm (Fig. 8g). The magnitude of the increase depends
on the local meteorological condition, the soil moisture as well as on the state of vegetation. Furthermore, in REMO2020-
iMOVE, evapotranspiration is composed of evaporation from bare soil, transpiration from vegetation, and evaporation from
skin reservoir. In the diurnal cycle (Fig. 8h), the evapotranspiration increase reaches its maximum at 1 PM LT, the hour with
the highest solar radiation in the model domain. During AMJ the increase in the evaporation of bare soil drives the changes
in evapotranspiration. Evaporation from the skin reservoir shows negligible effects, as it is only affected by the LAI and the
occurrence of precipitation or dew. Transpiration from vegetation shows a reduction through irrigation in comparison to the
not irrigated simulation from 9 AM LT to 4 PM LT in AMJ. Figure 8i shows the annual cycle of the effects on the different
evaporative fractions. The transpiration from vegetation is reduced with irrigation for March, April, and May before it shows an
increase from June to September of up to +14 mm month[-1]. The reduction in spring is explained by the slower development of
the LAI (Fig. 14a-b) in the irrigated simulation due to lower air temperatures (Fig. 11), which lead to reduced transpiration. In
different seasons of the year, different evaporative fractions are the driver of evapotranspiration (Fig. 8i). Bare soil evaporation
increases with irrigation and is the main driver of irrigation effects in evapotranspiration until July with the highest increase of
+28 mm month[-1] in April. Once the LAI reaches its maximum in July (Fig. 14a), it becomes the driver of evapotranspiration.
After the crops are harvested, there is only evaporation from bare soil and from the skin reservoir.





**Figure 8.** Effects on soil and surface processes on the irrigated fraction a,d,g) as spatial distribution of mean values of AMJ, b,e, h) as diurnal cycle of mean values irrigated areas in AMJ, and c, f, i) as annual cycle of mean values of irrigated areas for a-c) soil moisture, d-f) soil temperature in different depths, g-i) evapotranspiration fractions.

Irrigation affects the surface energy budget by changing the energy fluxes (Fig. 9). The latent heat flux increases by up to +150 W m$^{-2}$ and the sensible heat flux decreases by up to -120 W m$^{-2}$ in the Ebro Basin, in Sardinia, and in Laszio during April, May, and June. These changes lead to a shift in and a reduction of the Bowen ratio by up to -1 W m$^{-2}$ (Fig. 9a-c), which shows that the energy transfer between the surface and the atmosphere is driven by evaporative fluxes rather than sensible heat fluxes.

Irrigation effects on the surface energy balance in the irrigation hotspot regions show a diurnal cycle and are most pronounced during noon (Fig. 10). In SF, we see the strongest effects. There, irrigation increases the latent heat flux by up to +200 W m$^{-2}$,




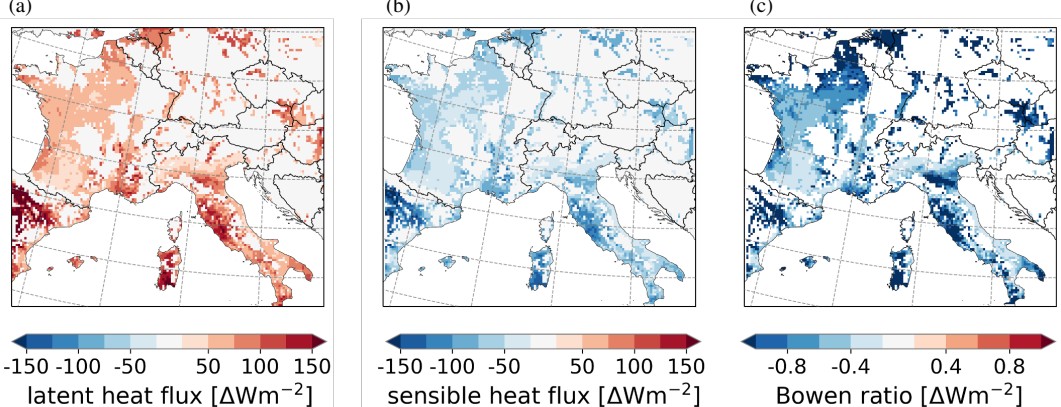

**Figure 9.** Effects on surface fluxes as spatial distribution of means for AMJ for a) latent heat flux, b) sensible heat flux, and c) Bowen ratio shift.

whereas it reduces the sensible heat flux by up to -185 W m$^{-2}$ during AMJ. The net radiation is slightly reduced in all three analysis regions which can be explained by a combination of a lower surface temperature (Fig. 8d), a reduced surface albedo due to higher soil moisture, together with increased humidity in the atmosphere and an altered cloud cover. The ground heat flux is calculated as residuum in the surface balance. During the irrigation hours, it decreases in all three analysis regions and

causes less heat storage in the ground.

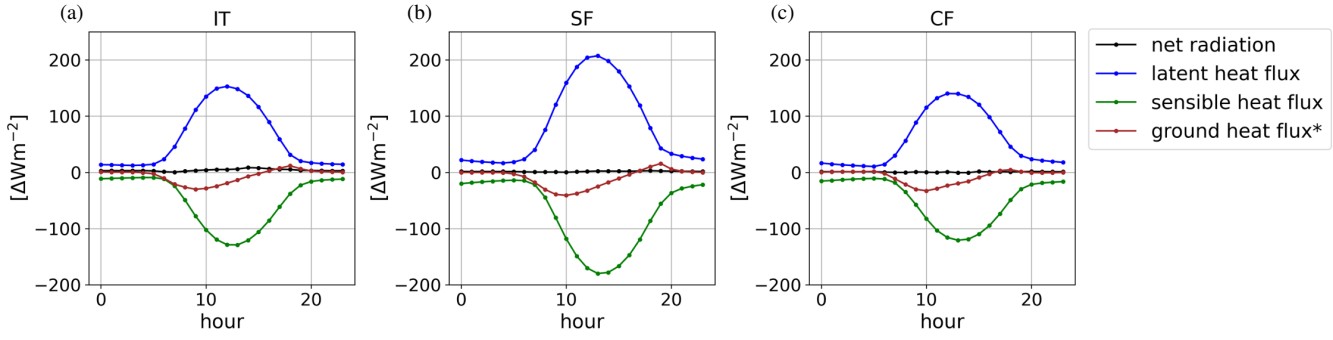

**Figure 10.** Effects on the surface energy balance of the irrigated fraction as hourly mean values of AMJ in the analysis regions a) IT, b) SF, and c) CF.

### 4.4.2   Effects on the atmosphere

Through land-atmosphere interactions, in particular through fluxes, the effects of irrigation propagate to the atmosphere. The effects occur mostly in grid cells with a high proportion of irrigated areas like in the Po Valley or the Ebro Basin (Fig. 11a and d). In both regions, the irrigation effects on the 2 m mean temperature (T2Mean) reach a reduction of up to -2 K averaged over



AMJ. Fig. 11b shows the diurnal cycle of T2Mean effects in the irrigated areas of the model domain. The whiskers and the outliers show the range of irrigation effects. Overall, T2Mean is reduced starting with the irrigation at 7 AM LT and reaching the highest reduction at 2 PM LT with about -3 K in irrigated areas. After that, the temperature reduction declines until the next irrigation starts at 7 AM LT the next day. We can find outliers showing a slight temperature increase, which is connected to grid cells with a low proportion of irrigated areas. Overall, the median shows a temperature reduction of -0.3 K in the irrigated

areas of the model domain.

In Fig. 11c, the monthly mean of the irrigation effect on the 2 m daily maximum (T2Max), minimum (T2Min), and T2Mean is shown. T2Max shows the strongest irrigation effects, whereas T2Min shows the smallest. The effects develop within the first irrigation month in March and reduce the 2 m temperatures. In the course of the year, the effects increase until irrigation stops in August, which is the first not completely irrigated month. As mean of the irrigated areas in the model domain, the

highest temperature reduction for T2Max and T2Mean is reached in July with -0.68 K and -0.39 K, respectively. In contrast, T2Min reaches its strongest temperature reduction in April with -0.21 K. With the end of irrigation the temperature reduction declines from August for the 2 m temperatures. T2Min reaches a temperature increase in the simulations with irrigation from September to November, which can be explained by the higher humidity in the atmosphere and its higher heat absorption as the driving effect. During the growing season, this effect is masked by the evaporative cooling from vegetation and soil. In May, the

temperature reduction declines due to the smaller irrigation requirement. The increases in the latent heat flux (Fig. 9b) and the evaporation (Fig. 8g-i) lead to an increase in the 2 m relative humidity (Fig. 11d-f). As for the 2 m temperature, the irrigation effects are in particular pronounced in grid cells with a high proportion of the irrigated fraction, as in the Po Valley and the Ebro Basin. The 2 m relative humidity increases in these grid cells by up to +20 % as mean for AMJ. Areas with smaller irrigated fractions reach a 2 m relative humidity increase by +8 %. This wide range of effects occurs also in the diurnal cycle, where

the strongest irrigation effects develop in the evening hours after the irrigation stops (Fig. 11e) and the air temperature starts to decrease. Then, the relative humidity increases by up to +23% in single grid cells. However, the median for the irrigation effect on 2 m relative humidity is at +3 %. In the annual cycle, the irrigation effect on 2 m relative humidity starts with irrigation in March. March and April as the first irrigated months reach the highest 2 m relative humidity increase through irrigation because these are the months with the highest irrigation requirement. In the course of the year, the irrigation effects decline to

a minimum in October with less than 1 % as spatial mean of the irrigated areas (Fig. 11f).

For precipitation, the effects of irrigation are not as clear as for the 2 m temperatures and 2 m relative humidity (Fig. 12). In the spatial distribution, there is no clear pattern of the irrigation effects (Fig. 12a). There are areas along the Alps in which precipitation increased by +100 mm as a monthly mean value for the AMJ. However, the pattern is very patchy. As monthly mean values for the whole model domain, the precipitation increases slightly during the irrigated months from March to July

(Fig. 12b and c). After irrigation stops in July, precipitation shows a reduction in comparison to the not irrigated simulation in August and September before it increases again from October to December. In our model setup, precipitation is represented with the shallow convection parameterization. To be able to analyze the physical processes that affect precipitation, we would have to resolve convection.



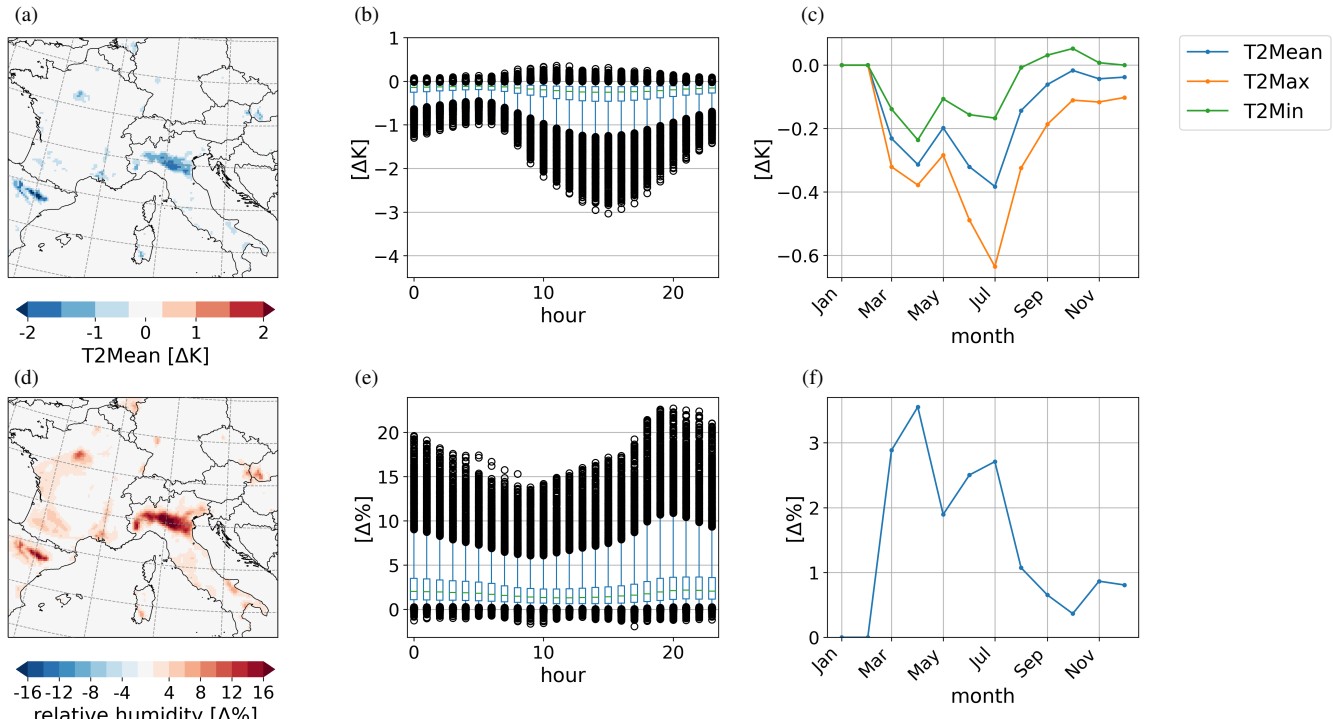

**Figure 11.** Effects on the atmosphere above irrigated areas as a and d) spatial distribution of mean values of AMJ, b and e) diurnal cycle of mean values in AMJ, and c and f) as annual cycle of mean values for a-c) 2 m temperatures and d-f) 2 m relative humidity.

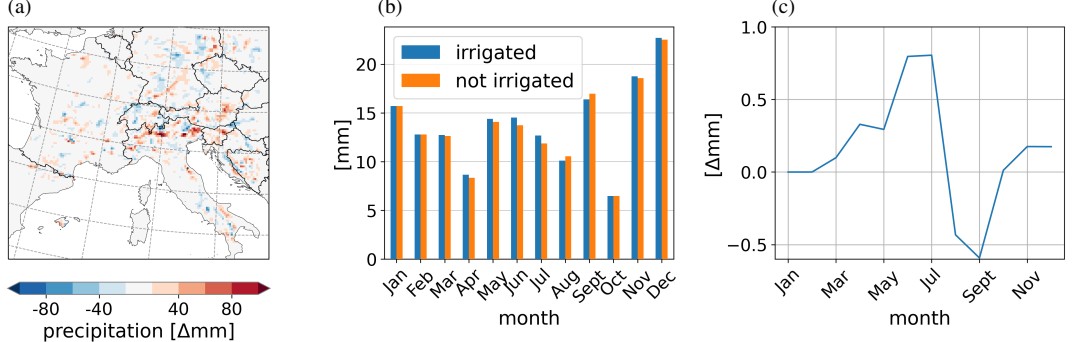

**Figure 12.** Effects on summed precipitation above irrigation areas as a) spatial distribution of mean values of AMJ, b) monthly mean values of the irrigated and not irrigated simulation, and c) as monthly mean effects.





### 4.4.3 Effects on the vegetation

For the vegetation modules of iMOVE, soil moisture is a crucial variable that drives multiple plant processes, such as the growing and shedding of leaves represented in the LAI. In addition, the LAI is driven by a growing degree threshold of temperature, simulating the growing season. Reaching the growing degree threshold, the LAI will decrease through harvest in the model. Due to the warm summer, the growing season ends in the southern parts of the model domain in Mid of July. As shown in Fig. 13a-c, the irrigation effects on the LAI depend on the month, in particular on the progressing growing season,

and on the region. In April and May (Fig. 13a and b), the LAI decreases in wide parts of the model domain such as Central France (Fig. 13b) by -1 $m^2m^{-2}$. This negative irrigation effect is caused by the 2 m temperature reduction (Fig. 11a-c), which is one of the drivers of LAI development leading to slower LAI growth in the first months of the growing season in the irrigated simulation (Fig. 14a and b). The more the growing season progresses and the vegetation approaches harvest, irrigation shows a positive effect on LAI. In June, the LAI increases with irrigation (Fig. 13c) in the Po Valley, the Ebro Basin, as well as

in Sardinia; areas that have experienced a warm summer and where the growing season is about to end. The LAI increases with irrigation because vegetation is never experiencing water stress. In June, the irrigation leads to smaller LAI in northern France as well as in parts of Germany. Again, the growing season has not yet progressed so far and the LAI develops slower with irrigation than without irrigation. The effects on the LAI drive mainly the effects on net primary production (NPP). In this study, NPP values refer to the carbon of fresh matter, following the description in Wilhelm et al. (2014). In April and

May (Fig. 13d and e), the irrigation effects on NPP are very small because the growing season has not yet progressed far and vegetation just started to develop. From May onwards, irrigation increases NPP by +800 $gCm^{-2}month^{-1}$ in the Ebro Basin as well as in the Po Valley. Where the LAI decreased (Fig. 13b), the NPP also decreases slightly, as in Central France. In June, the NPP increases through irrigation by up to +1200 $gCm^{-2}month^{-1}$. As in the LAI, the influence of irrigation on NPP is greater as the growing season progresses. The LAI and the NPP reach their maximum in June in both simulations, with and without

irrigation (Fig. 14a and c). The maximum irrigation effects of the LAI and NPP are reached shortly before the harvest in July (Fig. 14).

### 4.4.4 Delayed effects during a heat wave

As described in subsection 4.3, SW Europe, particularly Italy experienced a heat wave in early August 2017. Therefore, we will focus on the region IT including the Po Valley with its high fraction of irrigated areas for this analysis (Fig. 3). Due

to its temperature-reducing effect (Fig. 11a-c), irrigation is able to reduce the intensity of heat waves. In our experiment, irrigation is performed exclusively in the growing season. The growing season depends on the 2 m temperature. In 2017, the summer in IT was exceptionally warm and the growing season ended in July (Fig. 14a), thus, there was no irrigation during the August heat wave in IT (Fig. B1). Nevertheless, irrigation shows delayed effects. Even if there was no active irrigation, the 2 m temperature is reduced during the heat wave by previous irrigation. T2Mean is reduced by up to -4.5 K and T2Max

is reduced by up to -6.6 K. As with active irrigation (in AMJ, subsubsection 4.4.2c), the reduction of T2Min is smaller than for the maximum temperature and reaches -2.5 K in the north-western part of IT (Fig. 15b and c). Fig. 16a shows the 2 m



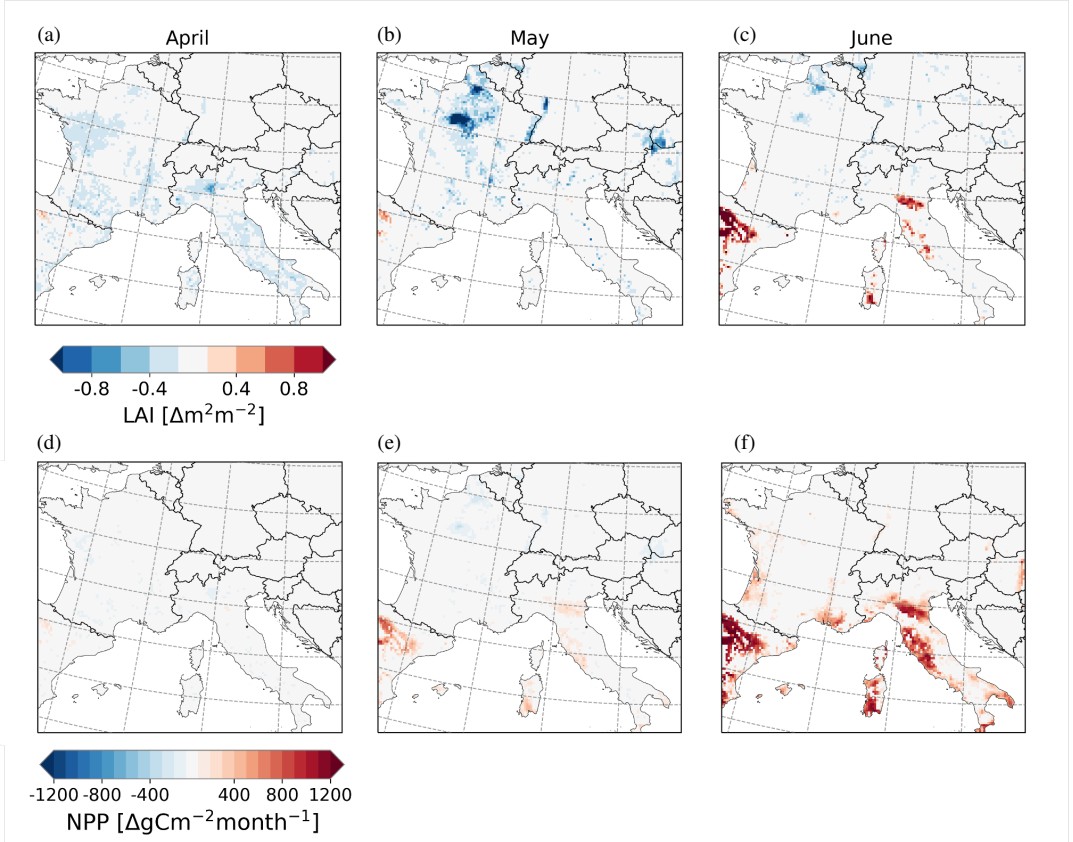

**Figure 13.** Effects on vegetation as spatial distribution of monthly mean values of the irrigated fraction, a, b and c) of LAI; d, e, and f) of NPP of cropland in Carbon of fresh matter.

temperature development during the week of the heat wave from the 1st of August until the 7th of August for IT. In both simulations the hottest days are the 3rd and 4th of August, however, T2Max is reduced by -1.5 K in the irrigated simulation reaching 35 °C instead of more than 36 °C. After the peak of the heat wave, the 2 m temperature drops from the 5th of August

in both simulations. In the irrigated simulation, the relative soil moisture stays close to saturation at a high level of 0.91 of wsmx after irrigation stopped, whereas in the non-irrigated simulation, it stays at a low level of 0.45 of wsmx(Fig. 16b). In IT, precipitation (Fig. 16c) occurs on the 2nd and 3rd of August at very low rates, which can be neglected, and on the 5th, 6th, and 7th of August at higher rates up to 4.5 mm day$^{-1}$ in the non-irrigated simulation and 2.5 mm day$^{-1}$ in the irrigated simulation. However, these precipitation rates are very low and affect the soil moisture with a small increase from 0.45 of wsmx to 0.47

of wsmx in the non-irrigated simulation on the 5th of August. During the heat wave, transpiration of the remained vegetation as well as evaporation of the soil are the drivers of evapotranspiration (Fig. 16d). However, in the irrigated simulation the evapotranspiration rate with up to 4 mm day$^{-1}$ is almost double as high as the evapotranspiration rate if irrigation is not turned on. This difference can be explained by the evaporation of bare soil. In the irrigated simulation the soil remained close to





**Figure 14.** Development of a) LAI and c) NPP, and the irrigation effects on b) LAI and d) NPP of the irrigated fraction.

saturation (Fig. 16b) and can evaporate. In the non-irrigated simulation, the soil moisture is at a very low level and is barely
evaporating (Fig. 16d). After the precipitation events, also the skin reservoir is evaporating on the 6th and 7th of August.
The delayed irrigation effects decrease the intensity of the heat wave and provide moisture in the soil to be evaporated which
can prevent the wilting of vegetation.

## 4.5  Comparison with observational data

The results of the simulations are compared to observational data collected within SCIA (subsection 2.3). For the comparison,
we have focused exclusively on the Po Valley, represented in the analysis region IT (Fig. 3). In the Po Valley, we have the
largest cluster of grid cells with a high proportion of irrigated fraction (Fig. 3) and therefore, the most developed irrigation ef-
fects in the atmosphere. To compare the model results to the observational data, we filtered the SCIA data for April to August,
as months with active irrigation, as well as months with delayed irrigation effects. Further, we filtered the SCIA data for the
location in the IT region and the presence of irrigated fraction. We selected the SCIA data with a higher irrigated fraction than





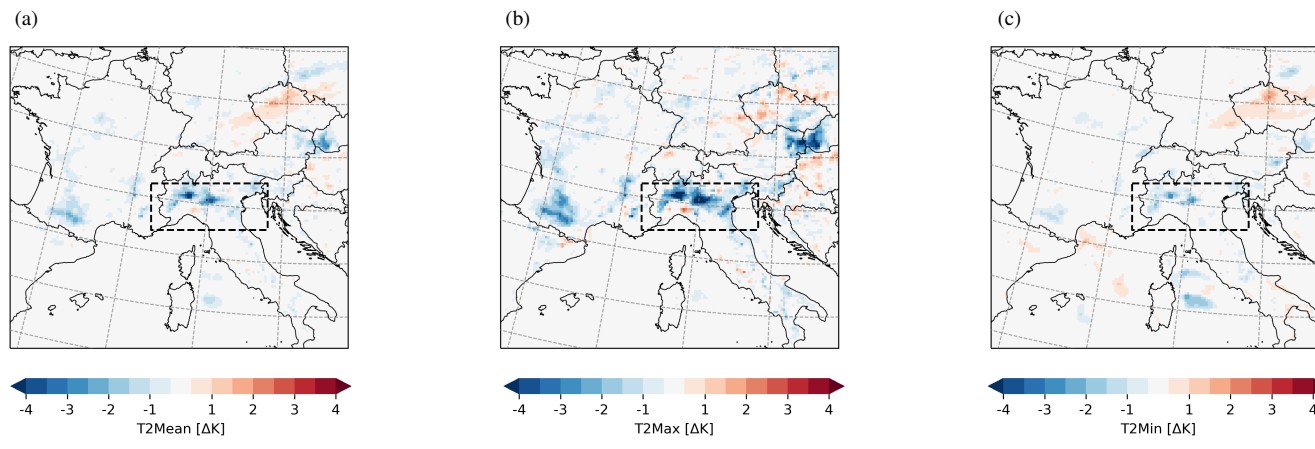

**Figure 15.** Delayed irrigation effects on 2 m temperature during the heat wave from 01/08/2017-07/08/2017 in IT a) T2Mean, b) T2Max and, c) T2Min.

31 % which is the mean of irrigated fraction in that area to reach a clear signal from the irrigation effects. The model data were then interpolated to the locations of the filtered observational data using inverse distance weighting with four known points. We calculated the bias for each station location and averaged it across all locations for each month. As the last step, the statistical significance of the bias distributions is evaluated with a student's t-test for two independent samples using a significance level ($\alpha$) of 0.05. This process was performed for the results from the simulation with irrigation as well as for the results from the

simulation without irrigation. The filtering results in a different number of suitable station data for each variable (Table 3, Fig. C1).

For this comparison, we focus on the near-surface temperature variables T2Mean, T2Max, and T2Min. In general, the irrigation parameterization reduces the 2 m temperatures. Without irrigation, REMO2020-iMOVE overestimates T2Mean from

April to August in IT. Using the irrigation parameterization, the bias can be significantly reduced from April to July, in particular in May with a remaining bias of 0.04 K. However, July and especially August have the largest bias in the irrigated and not irrigated simulation results. The delayed irrigation effects cause only a minor, non-significant bias reduction in August from 4.67 K in the not irrigated simulation to 4.47 K in the irrigation simulation. The large biases in July (irrigated: 1.41 K, not irrigated: 3.36 K) and August are most probably connected to the early harvest and the drop in vegetation. Vegetation is an

important contributor to the evapotranspiration of the surface which has a cooling effect on the 2 m temperature (Fig. 8g-i). The early harvest and the early end of the growing season lead to an end of active irrigation.

For T2Mean, the irrigation parameterization caused significant bias reductions from April to July with high t-values and p-values of 0.0. For T2Max and T2Min, the results are not as clear as for T2Mean. REMO2020-iMOVE overestimates T2Max in April, June, July, and August. Using the irrigation parameterization leads to an underestimation of T2Max, except for August

when the delayed cooling effect of irrigation reduces the large bias of 4.61 K to 3.65 K. Again, August has the largest bias





**Figure 16.** Development of delayed irrigation effects during the heat wave in August (01/08/2017-07/08/2017) in IT as a) spatial mean of 2 m temperatures, b) spatial mean of relative soil moisture c) spatial sum of precipitation, f) spatial mean of evapotranspiration.

in both simulations and can be explained by the drop of vegetation. In general, T2Max is represented closer to observational values without irrigation.

The T2Min is overestimated with and without the irrigation parameterization by REMO2020-iMOVE. However, the irrigation parameterization significantly reduces the bias in April, June, and July. As for T2Mean and T2Max, August is the month with the largest bias in both simulations. However, the irrigation parameterization increases the bias even more from 5.47 K to 5.89




K this time with its warming effect in August for T2Min (Fig. 11). The results for T2Min show lower t-values and larger p-values pointing out the lower robustness of the bias distributions.

**Table 3.** 2 m temperature bias for the irrigated and not irrigated simulation with t-test results. Bold values indicate statistical significance with $\alpha$=0.05.

|  | **T2Mean** (51 stations) | | | | **T2Max** (60 stations) | | | | **T2Min** (53 stations) | | | |
|---|---|---|---|---|---|---|---|---|---|---|---|---|
|  | **irri** | **noirri** | **t-value** | **p-value** | **irri** | **noirri** | **t-value** | **p-value** | **irri** | **noirri** | **t-value** | **p-value** |
| April | **0.20** | 1.37 | -7.43 | 0.0 | -1.36 | 0.43 | -10.6 | 0.0 | **2.03** | 2.55 | -2.0 | 0.05 |
| May | **0.04** | 0.98 | -5.02 | 0.0 | -1.55 | -0.03 | -9.17 | 0.0 | 1.68 | 2.05 | -1.41 | 0.16 |
| June | **-0.11** | 1.44 | -7.3 | 0.0 | -2.39 | 0.24 | -12.09 | 0.0 | **2.14** | 2.7 | -2.11 | 0.04 |
| July | **1.41** | 3.36 | -10.21 | 0.0 | **-0.34** | 2.98 | -16.61 | 0.0 | **3.07** | 3.99 | -3.52 | 0.0 |
| August | 4.47 | 4.67 | -1.3 | 0.2 | **3.65** | 4.62 | -6.5 | 0.0 | 5.89 | 5.47 | 1.9 | 0.06 |

## 5 Discussion

We developed a new subgrid parameterization representing channel irrigation and implemented it in the regional climate model system REMO2020-iMOVE. An older version of the model, REMO2009, was tested with an irrigation parameterization by Saeed et al. (2009) before. The study analyzed large-scale irrigation effects over the Indian subcontinent on 0.5° horizontal resolution. In contrast to our study, the parameterization represented irrigation in the whole model grid cell leading to possible overestimation of irrigation effects. However, it pointed out the importance of representing irrigation in climate models, in particular over large-scale, intensely irrigated areas such as the Indus Basin, because irrigation decreases dry biases and affects the development of meteorological patterns such as the South Asian Summer Monsoon by adding water to the climate system (Saeed et al., 2009). In our experiment, we focus on higher-resolution simulations. The representation of irrigation on subgrid scale is an improvement in the representation of irrigated areas and qualifies the parameterization for high-resolution studies in heterogeneous regions such as Europe. According to Im et al. (2010) and Giorgi and Avissar (1997), subgrid-scale representation of land cover and land use improves the representation of land-atmosphere interaction in climate models. In the new parameterization, irrigation is exclusively realized, where it is required. Therefore, only the irrigated fraction is part of the irrigation process. The subgrid-scale approach is also used in e.g. Lawrence et al. (2019) and Ozdogan et al. (2010). Our irrigation parameterization has different water application schemes that can be used to address different research questions. An influence of the different water application schemes on irrigation effects could not be found for similar settings. However, it has to be considered that the irrigation effects depend strongly on the irrigation amount, which in turn depends on the soil hydrology of the climate model. Due to the bucket scheme in REMO2020-iMOVE, suitable prescribed values of the irrigation amount differ from observed values because the water is added to the whole soil column. Therefore, model-specific values need to be chosen and the irrigation amount cannot be validated with observational values. However, to simulate irrigation effects, irrigation using physical thresholds as irrigation start and target as in the flextime or adaptive water application scheme





are more suitable and are able to represent realistic soil conditions. For the future, for irrigation studies, we recommend the
representation of the soil hydrology with a multiple layer scheme as for WRF (Valmassoi et al., 2019) or CLM (Lawrence
et al., 2019; Ozdogan et al., 2010) it already exists, and which is currently under development for REMO2015 (Abel et al.,
2020; Rai et al., 2022). A multiple layer scheme will allow the usage of observed values for the irrigation amount and improves
the representation of soil hydrology. Further, to represent an observed irrigation amount, irrigation water loss through, e.g.,
evaporation or leaks during the water transport has to be considered. Our irrigation parameterization adds the irrigation water
directly to the soil moisture and therefore, is not taking into account irrigation efficiency.

For REMO2020-iMOVE and its bucket scheme, we selected the adaptive water application scheme as the default scheme
because it does not require model-specific values and reaches the irrigation target in the prescribed time. To the author's
knowledge, a nonlinear approach, such as the adaptive scheme, has never been used for irrigation parameterization before but
has proven to be suitable in this study.

The simulated effects of the irrigation parameterization on the surface energy balance are more pronounced in our study than in
comparable studies (Valmassoi et al., 2020b; Lobell et al., 2009). This can be explained by the newly implemented fraction that
has its own surface energy balance. In this study, we analyzed exclusively the values of the irrigated fraction and no grid cell
averages of the soil and surface variables. Further, in our experiment, we used the maximal irrigation target and the maximal
irrigation threshold to show the maximal possible effects. The effects on the atmosphere are in the same range as other studies
(Valmassoi et al., 2020b; Lobell et al., 2009; Thiery et al., 2017). For example, Valmassoi et al. (2020b) found a monthly
T2Max reduction of up to -3 K in the Po Valley whereas we found a T2Max reduction of up to -4 K in single grid cells. The
effect on vegetation, slowing down the development of the LAI is model specific and could not be verified with other studies.
In contrast, studies found that with irrigation the LAI is larger than without (Patanè, 2011). Therefore, the interactive LAI
representation in REMO2020-iMOVE might have to be improved. Further, the large positive bias in August in comparison to
observational data can be attributed to the missing vegetation and the early harvest in July, which is represented as LAI drop
causing a stop to vegetation processes. The missing evaporative cooling of the transpiration of vegetation leads to increasing 2
m temperatures. This effect was already observed by Wilhelm et al. (2014). Nevertheless, the irrigation parameterization could
significantly reduce the bias for T2Mean in 2017 in the Po Valley, particularly in months with active irrigation. For T2Max, the
irrigation parameterization adds a cold bias, whereas, for T2Min, the irrigation parameterization reduces the warm bias. We
can follow that the irrigation parameterization decreases the diurnal range of the 2 m temperature. However, as the warm bias
in T2Min is still high also with irrigation, other processes in the model need to be considered as the source. The underestima-
tion of T2Max can be traced back to our experiment design which shows maximum irrigation effects. And therefore, it might
overestimate irrigation effects. First, our irrigated fraction is based on the area equipped for irrigation that is not completely
irrigated in reality. And second, in our experiments, we keep the soil moisture at very high levels (higher than 0.75 of wsmx)
at which plants do not experience any water stress and the potential transpiration by plants is reached.

For our irrigation parameterization, we assumed unlimited water availability for all grid cells. However, for irrigation practice,
this is not the case. First, because the probability of heat waves and droughts in Western and Southern Europe increases with
climate change (Kew et al., 2019) and there is likely not sufficient water available during these periods (IPCC, 2019). Second,





during heat waves and droughts, politics have to ration the water as happened during the intense heat wave in 2022 in Northern
Italy (Balmer and Amante, 2022; Giuffrida, 2022). Having a limited water reservoir in REMO2020-iMOVE would be a step
towards a more realistic irrigation amount.

Our parameterization increases the soil moisture directly and therefore can be understood as a representation of channel irri-
gation. Additionally, there are more irrigation methods e.g. sprinkler or drip which require canopy interactions and different
parameterization approaches as Valmassoi et al. (2019) and Yao et al. (2022) pointed out.

## 6 Conclusions

By implementing irrigation into the regional climate model system REMO2020-iMOVE, we include a widely used land use
practice and an important aspect of anthropogenic forcing on the climate system, enabling the investigation of irrigation effects.
Our newly developed parameterization is designed for high-resolution studies using a separate irrigated land fraction ensur-
ing that exclusively irrigated areas are irrigated in the model and irrigation effects can be realistically estimated. Further, our
parameterization takes into account vegetation processes. With our model system REMO2020-iMOVE, we could show the ir-
rigation effects and feedbacks regarding LAI development, which develops slower in the model, but reaches higher maxima, or
regarding the process of NPP, which increases with irrigation. Our parameterization is characterized by three water application
schemes, which simulate irrigation with prescribed irrigation, with flexible time irrigation, and with adaptive irrigation. Even
though the irrigation schemes differ in irrigation time, irrigation events, and water application per timestep, the differences in
the effects are small and can be neglected. However, the different irrigation schemes can be applied to different research ques-
tions in the future. Rather than the water application, the water amount is an important driver of irrigation effects. Therefore,
simulations with a realistic irrigation amount together with a layer model are desirable for the future.

We applied our irrigation parameterization for dry and hot conditions in 2017 in SW Europe. Whereas the effects on soil and
surface variables are more pronounced in our study using the fractional approach than in comparable studies, the effects on the
atmosphere match the range of temperature reduction. For effects on small-scale precipitation, the resolution of our study is
not high enough and we cannot resolve convective processes, leading to unclear irrigation effects. Therefore, studies on higher
resolution, such as on convection-permitting scale, are necessary. For REMO2020-iMOVE the application of our irrigation pa-
rameterization decreased significantly the monthly warm bias of T2Mean during AMJ with active irrigation. But also delayed
irrigation effects occur, influencing the summer season. Our study showed that irrigation effects such as temperature reduction
and soil moisture increase are not only an adaptation measure during droughts or heat waves, these irrigation effects have the
potential to prevent or mitigate such climate extremes on local scale.

*Code and data availability.* The model code of REMO2020-iMOVE with the new irrigation parameterization is available on request (contact@remo-
rcm.de). The scripts used to produce the results presented in this paper are archived on Zenodo (https://doi.org/10.5281/zenodo.7889385), as



is the input data (https://doi.org/10.5281/zenodo.7867329). The code can be cited as Asmus and Buntemeyer (2023). The observational data
from SCIA can be downloaded from http://www.scia.isprambiente.it.

## Appendix A: Parameterization development

## A1    Implementing irrigation into REMO2020-iMOVE

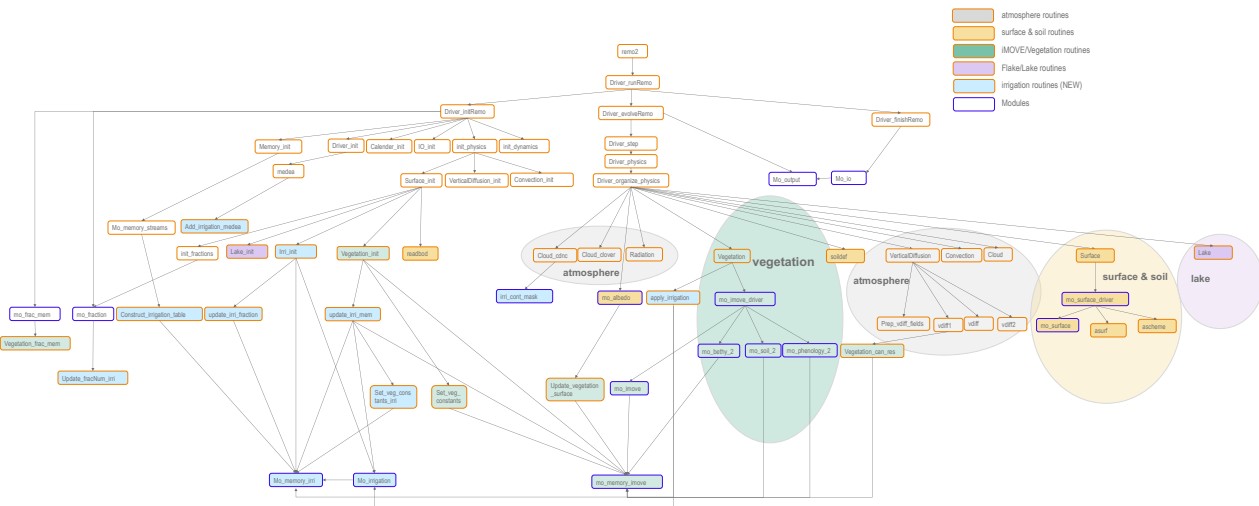

**Figure A1.** Remo2020-iMOVE+FLake call tree for version with subgrid irrigation.



## A2 Water application schemes

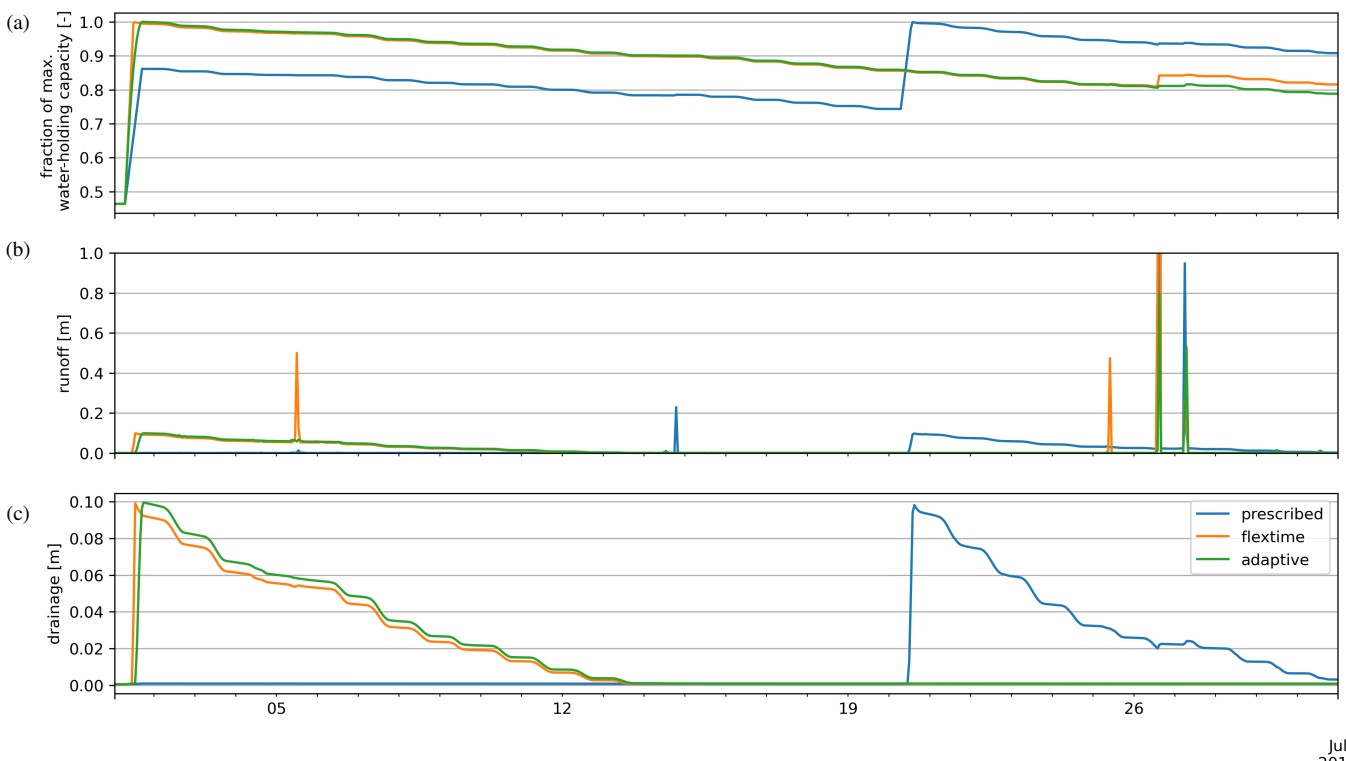

**Figure A2.** Results of different water application schemes (T1, T2, T3) for a) relative soil moisture, b) runoff, and c) drainage.



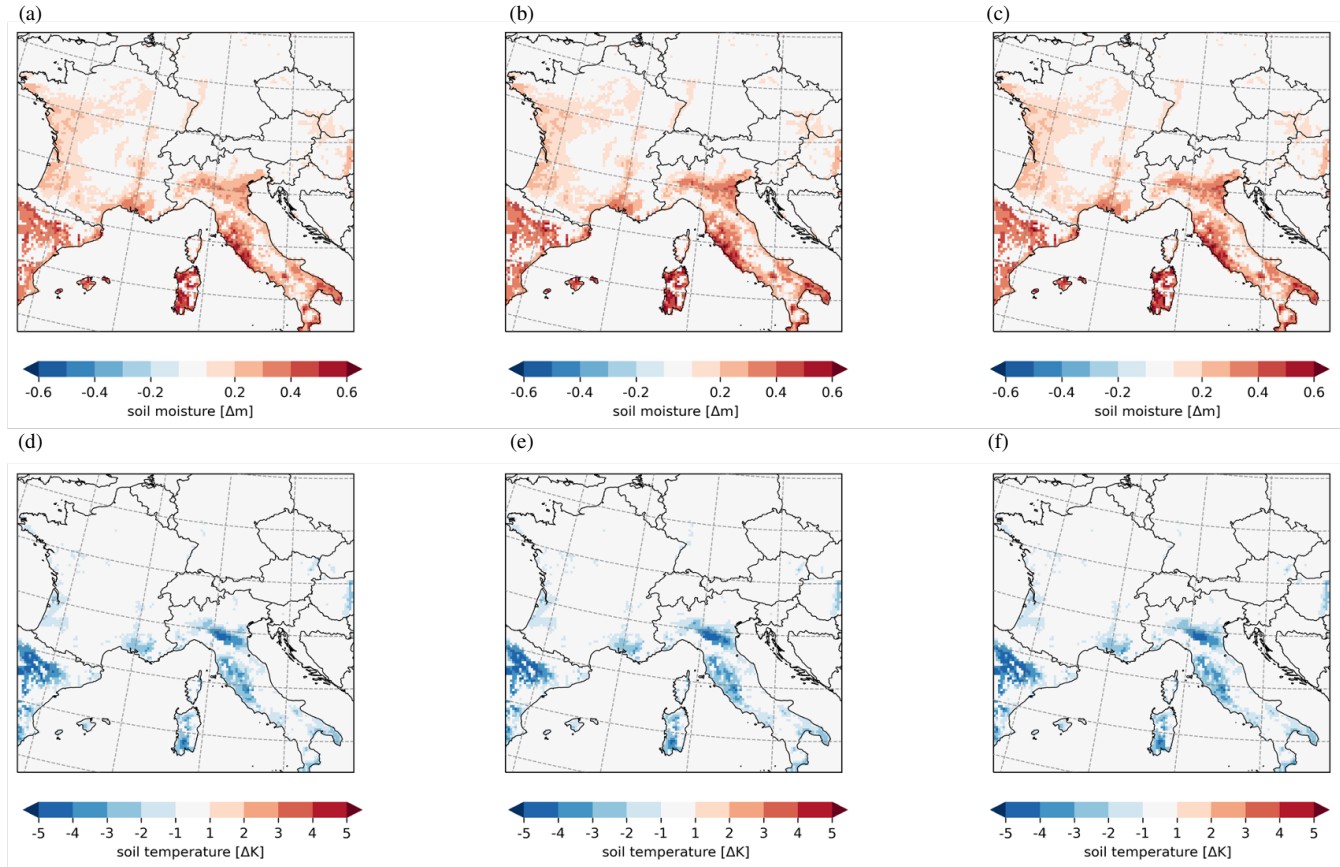

**Figure A3.** Spatial distribution of mean effects of different water application schemes in June 2017 for a) - c) soil moisture, d) - f) surface temperature using a) and d) prescribed, b) and e) flextime, c) and f) adaptive scheme



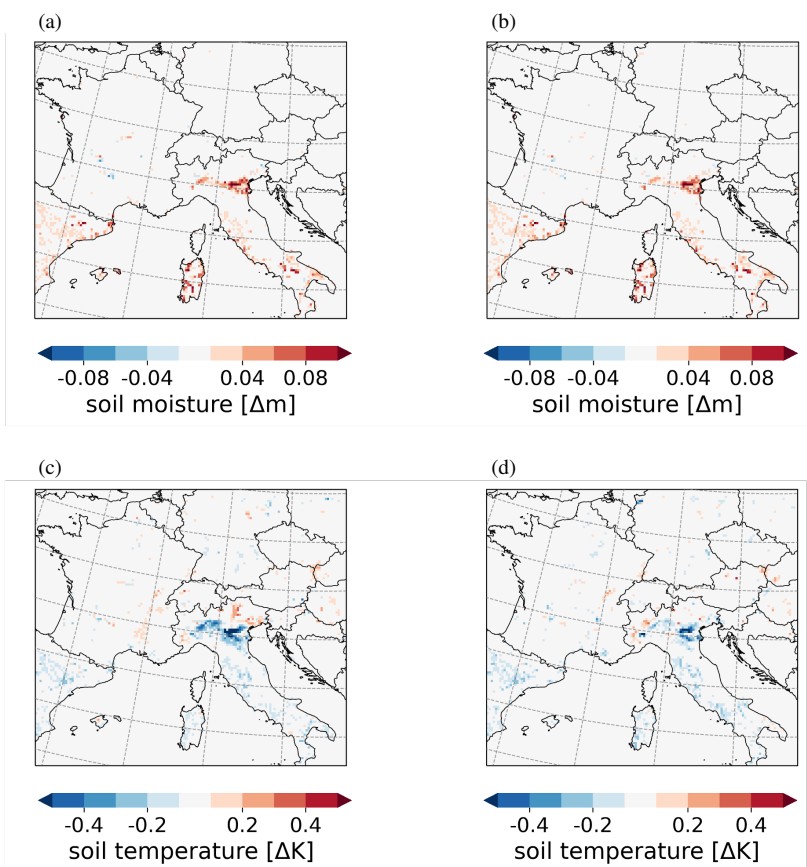

**Figure A4.** Differences between water application schemes in June 2017 for a) and b) soil moisture, c) - d) surface temperature between a and c) flextime-prescribed schemes, b and d) adaptive and prescribed scheme.



## Appendix B: Irrigation conditions during August

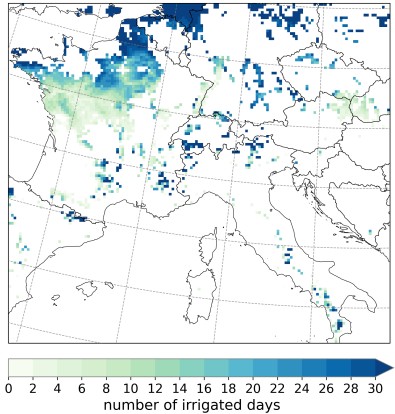

**Figure B1.** Number of irrigation days in August 2017.





**Appendix C: Station location used for comparison with observational data**

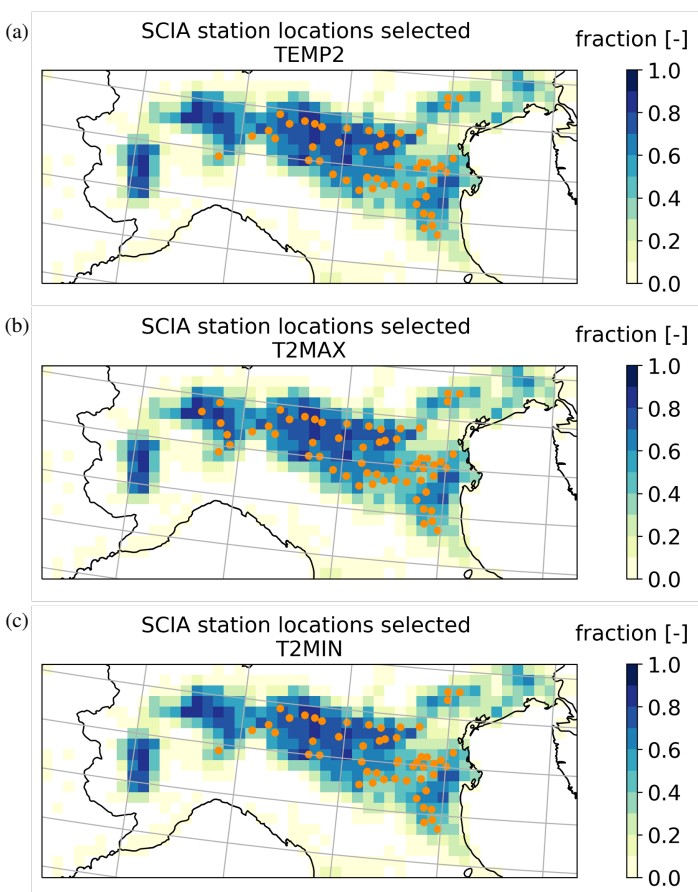

**Figure C1.** Station location for a) T2Mean, b) T2Max, and c) T2Min.

*Author contributions.* CA, PH, DR, and JB developed the experiments. CA developed the irrigation module. CA and JPP implemented the parameterization in the model code in close cooperation with PH. CA conducted the analysis incl. the visualizations under the supervision of DR and JB. CA prepared the initial manuscript. All authors reviewed the paper draft and contributed to the final paper.

*Competing interests.* The authors declare that they have no conflict of interest.

*Acknowledgements.* We are grateful for the support and help of the REMO developer team located at GERICS. Here, we want to thank in particular Lars Buntemeyer for preparing the ERA5 data as model forcing and helping with the publication of the analysis code and data. We



also thank Juliane El Zohbi for internally reviewing our paper draft. We are grateful to DKRZ for providing the high computing capacity, with which we performed our simulations. Further, we want to thank ISPRA for collecting and providing observation data publicly in the SCIA database, and FAO for providing the GMIA V5 publicly.

This work was financed within the framework of the Helmholtz Institute for Climate Service Science (HICSS), a cooperation between the Climate Service Center Germany (GERICS) and Universität Hamburg, Germany, and conducted as part of the LANDMATE (Modelling human LAND surface modifications and its feedbacks on local and regional cliMATE) project.



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
