# Peer review of "Modeling and evaluating the effects of irrigation on land-atmosphere interaction in South-West Europe with the regional climate model REMO2020-iMOVE using a newly developed parameterization"

_EGUsphere, 2023_

## Author Response (AR1)

**Answers to the comments by the authors**

| line | comment from | comments from referee/community/executive editor | author's response | author's changes in manuscript (in bold) with line number in revised version |
|---|---|---|---|---|
| - | Jozsef Szilagyi | I wonder if the authors are familiar with the findings of "Anthropogenic hydrometeorological changes at a regional scale: Observed irrigation-precipitation feedback (1979-2015) in Nebraska, USA" by Szilagyi and Franz in Sustainable Water Resources Management, 6(1), 1-10. It would be interesting to see if the model can in theory reproduce this observation of local precipitation suppression by regional scale irrigation! | Dear Joszsef, thank you for your valuable hint. The results of your study are very interesting. We were not aware of it, but included it now in our paper. In our study, irrigation increases the monthly precipitation in all months except for August and September (Fig. 12). However, we simulated only one year. To be able to make clearer statements about precipitation-irrigation feedback a longer experiment and convection-permitting resolution are needed. Convective precipitation genesis depends on multiple, interconnected factors such as atmospheric stability, moisture content, temperature profiles, and wind patterns. The climatology and surrounding meteorological conditions are further influencing factors, which drive the develpment of irrigation effects. Our region is characterized by a Mediterranean climate as well as showing influences from the Alps, whereas Nebraska shows a continental climate. These regional differences are also shown eg. in Thierry et al. 2017, Lobell et. al 2009. | line 570: **In our study irrigation effects on precipitation remain unclear and cannot reproduce the findings in observation studies showing a regional annual decrease of precipitation as found in Szilagyi and Franz (2020). However, irrigation effects on precipitation are indirect and influenced by many interconnected factors such as atmospheric stability, specific humidity, temperature, and wind patterns. These make a comparison difficult. To find clearer patterns of irrigation effects on precipitation, a longer experiment is necessary. Further, in our study, the convective precipitation is parameterized and the generating processes are not resolved. Therefore, we recommend using convection-permitting resolution for analyzing precipitation-irrigation feedback.** |
| 170-171 | Referee 1 | Do this account for soil irrigation history? I mean, if you initialize the simulation the 15th, it accounts for the irrigation on the 14th? | I am not sure if I understand your question. In line 170 we describe the order of the physical processes called in every timestep in REMO2020-iMOVE. During the initialization, there is no irrigation carried out. The irrigation is only defining and adding the new irrigated fraction to the model. We added more details in the text to clarify the call structure and we give more details of the soil initialization following your comment to line 308 (row 9 in this table). | line 169: **The irrigation module, which accounts for a check for irrigation requirement and the water application, is called every time step exclusively for the irrigated land fraction. These irrigation processes are carried out after the hydrological processes of the soil from the previous time step (t-1). In this way, the irrigation processes are applied to the soil hydrology inherited from t-1. After the irrigation processes, the vegetation processes start, which are strongly influenced by the moisture content in the soil and in the atmosphere of the same time step (t).** |
| whole text | Referee 1 | I would suggest transform the abbreviations in lowercase in italics, in order to differentiate them from the normal text. Ex: line 176 irrht, line 195 wsmx | Thank you for your suggestions. We applied it. However, we kept the names of the analysis regions in capital letters, since these are rather names than abbreviations and are defined in Fig. 3. | in whole text, eg., line 179: threshold (*irrthr*),  line 118: maximum water-holding capacity (***wsmx***), line 119: land surface parameters (***LSP***) |
| 196-197 | Referee 1 | Why is this relaxation applied? | Since this approach of our parameterization is based on a soil moisture target rather than on an absolute irrigation water amount as in the prescribed scheme, we have to take into account the soil moisture changes not related to irrigation during the irrigation time. The relaxation approach allows us to reach our target value slowly within the prescribed time simulating the increasing soil moisture during irrigation process. We improved our text and pointed out the advantages. | line 199: The "adaptive irrigation" is also based on a prescribed soil moisture target and a prescribed, limited irrigation time. Again, for each grid cell, the water amount is calculated that is necessary to reach the soil moisture target. The water amount added every time step follows a relaxation approach (Eq. 1), **which simulates the increase of soil moisture during the time steps of irrigation and simultaneously, considers the changes of soil moisture not related to irrigation. Further, our relaxation approach takes into account the number of irrigation time steps remaining. Using this approach the soil moisture increases until the irrigation target is exactly reached during the prescribed irrigation time.** |
| 288 | Referee 1 | please, define again the regions here, to make it easy the reading. | Thank you for your comment. We applied it. | line 298: Fig. 7 shows the evolution of the meteorological conditions from April until August in the three analysis regions, **IT ( Italy), CF (Central France), and SF (Spain-southern France)** (Fig. 3). |
| 308 | Referee 1 | Is the soil moisture initialized by the reanalysis? Orthere is a surface model that is computed offline? | The initialization of the experiments is explained in section 4.1 Experiment Setup. Therefore, we improved our description here. To clarify: Soil conditions in RCMs need a long spin-up time until they are in equilibrium state. Therefore, for the initialization of the model, the soil variables from a previous, > 10 year long REMO simulation for the the same date as the starting date of the simulation (01/01/2017) are used. Except for the soil variables, the initialization of all other variables is based on reanalysis data (ERA5) (see Table 2). This is a common method in regional climate modeling to safe computation time. After the initialization, soil moisture is calculated by REMO2020-iMOVE, including evapo(transpi-)ration, infiltration, runoff, drainage, and for the irrigated simulations irrigation. We improved our text and tried to clarify. | line 219: S0 and S1 **start from 01/01/2017. We initialized S0 and S1 with ERA5 (Table 2), except for the soil conditions. Since soil conditions have a long spin-up time in RCMs, we initialize the soil variables with a previous, long-term (> 10 years) REMO simulation to get the soil variables in an equilibrium state. This method is also known as "warm-start" ( Pietikäinen et al., 2018).** |
| 309-310 | Referee 1 | why is the irrigation applied during the day hours? I though irrigation is typically done in the evening, night or or early morning. Do you have data to reference the applied daytime hours of irrigation? | Thank you for your comment. The timing of irrigation depends on the irrigation method, the region and on the plant. We base our assumption of irrigation techniques and conditions based on findings from irrigation in the Po Valley, since it is the largest irrigated area in my model domain. There, channel irrigation (furrow irrigation) is one of the most common irrigation techniques which can take up to 24 h (Bjorneberg, 2013, Zucaro et al. 2014) untill the channels are filled with sufficient water.In addition, in the Po Valley, irrigation is politically regulated with a rotational supply in predetermined shifts to its users (Zucaro et al. 2014). Therefore, irrigation is possible/necessary during daytime hours. We also found the same irrigation start time in Valmassoi et al. 2019 who investigated the Po Valley as well and tested different starting times. We improved our references and the descriprion of irrigation details in the text in subsection 3.2 Irrigation module and water application schemes, as well as in 4.1. Experiment setup and in 5 Discussion. | line 178: Using an adjustable threshold (irrthr) on the soil moisture, the irrigation module determines the grid cells with irrigation requirements and creates a daily irrigation mask **at 7 AM LT as the starting time for irrigation in our parameterization following Valmassoi et al. (2019) .**
line 231:  Following Bjorneberg (2013) **and Zucaro (2014)** channel irrigation is performed **for up to 24 hours depending on the channel width and length, we chose 10 h irrigation time for our experiment. With the irrigation start time at 7 AM LT (subsection 3.2), irrigation is applied during daytime in our experiment.**
line 554: The underestimation of T2Max can be traced back to our experiment design which shows maximum irrigation effects. And therefore, it might overestimate irrigation effects. First, our irrigated fraction is based on the area equipped for irrigation that is not completely irrigated in reality. Second, in our experiments, we keep the soil moisture at very high levels (higher than 0.75 of wsmx) at which plants do not experience any water stress and the potential transpiration by plants is reached. **And third, we irrigate at daytime hours leading to strong effects on variables with a distinct diurnal cycle as the surface fluxes, evapotranspiration as well as T2Max. The effect of irrigation timing was analyzed by Valmassoi et al. (2019), who showed a rather low impact of irrigation timing on the development of irrigation effects.** |
| 319 | Referee 1 | "The strongest cooling effect in the soil occurs…" | We applied your more defined suggestion. | line 330: The strongest cooling effect **in the soil** occurs in the Ebro Basin and in the southern Po Valley with -4 K as a mean value in AMJ. |
| Figure 8 caption. | Referee 1 | Figure 8 caption. Please, explain it better, what is delta_m? | Thank you for pointing out the unclarity. We changed the label of the y-axes and legends and to keep only the unit in square brackets in all figures showing irrigation effects as differences. | New Fig. 8, Fig. 9, Fig. 10, Fig. 11, Fig. 12, Fig. 13, Fig. 14, Fig. 15, |
| 356 | Referee 1 | Bowen ration does not have units | Thank you for pointing out this mistake. We corrected it. | line 367: These changes lead to a shift in and a reduction of the Bowen ratio by up to -1  (Fig. 9a-c), which ... |
| Figure 9 | Referee 1 | Figure 9: again Bowen ration does not have units | Thank you for your pointing out this mistake. We corrected it. | New Fig. 9 |

| line | comment from | comments from referee/community/executive editor | author's response | author's changes in manuscript (in bold) with line number in revised version |
|---|---|---|---|---|
| Figure 9 caption | Referee 1 | Figure 9 captions: Instead of "Effects on surface..etc" please, explain what you compare, what is delta, etc. | Thank you for pointing out the unclarity. We improved the captions for all figures showing irrigation effects. | Figure 8. **Irrigation effects based on the difference between the simulation with irrigation (S1) and the simulation without irrigation (S0)** on soil and surface processes .... Figure 9. **Irrigation effects based on the difference between the simulation with irrigation (S1) and the simulation without irrigation (S0)** on surface fluxes ... Figure 10. **Irrigation effects based on the difference between the simulation with irrigation (S1) and the simulation without irrigation (S0)** on the surface energy balance ... Figure 11. **Irrigation effects based on the difference between the simulation with irrigation (S1) and the simulation without irrigation (S0)** on the atmosphere ... Figure 12. **Irrigation effects based on the difference between the simulation with irrigation (S1) and the simulation without irrigation (S0)** on summed precipitation ... Figure 13. **Irrigation effects based on the difference between the simulation with irrigation (S1) and the simulation without irrigation (S0)** on vegetation ... Figure 14. Development of a) LAI and c) NPP, and the **irrigation effects based on the difference between the simulation with irrigation (S1) and the simulation without irrigation (S0)** ... Figure 15. Delayed **irrigation effects based on the difference between the simulation with irrigation (S1) and the simulation without irrigation (S0)** on 2 m temperature ... Figure 16. Development of delayed **irrigation effects based on the difference between the simulation with irrigation (S1) and the simulation without irrigation (S0)** during the heat wave ... |
| 367 | Referee 1 | First sentence should be rewritten: "The effects of irrigation propagate to the atmosphere through land-atmosphere interactions, in particular through fluxes." | We applied your suggestion. | line 378: **The effects of irrigation propagate to the atmosphere through land-atmosphere interactions, in particular through fluxes.** |
| Figure 11 | Referee 1 | 11: could you plot outliers less thick? | Thank you for your suggestion. Less thick was not possible, but we selected another marker and changed the color and transparency for the outliers. In addition we increased the line width of the box and whiskers to make clear what is important. We believe, these changes improved the readability of the diagram. | New Fig. 11b and e) |
| Figure 11 caption | Referee 1 | Fig 11 caption. Please, explain what limits indicate the boxplots in the box, whiskers and outliers. | We applied your suggestion. | Figure 11. Irrigation effects based on the difference between the simulation with irrigation (S1) and the simulation without irrigation (S0) on the atmosphere above irrigated areas as a and d) spatial distribution of mean values of AMJ, b and e) **mean diurnal cycle in AMJ with the box spanning from the 1st to the 3rd quartile, the red line showing the median, the whiskers showing the 5th and 95th percentile, and outliers as values outside of these limits**, and c and f) as annual cycle of mean values for a-c) 2 m temperatures and d-f) 2 m relative humidity. |
| 443 | Referee 1 | Why do you think there is the decrease in the daily accumulated precipitation in the irrigated simulation? Please, give an hypothesis, or a hint. | Referring to Fig. 16c, we show only specific dates of the simulations. For the 05/08/ precipitation increases through irrigation, for the 06/08/ decreaes and for the 07/08/ showing no effect for our analysis region IT during the heat wave. As showing also in Fig. 12, the effects of irrigation on precipitation are not clear in our simulations. For the 06/08/ we can only give a hypothesis. Irrigation does not only increase the moisture in the soil and atmosphere, through changes in the temperatures it has the potential to also affect the wind conditions. Analysing limiting areas gives the option, that the increased humidity from the evapotranspiration in the irrigated simulation could have been advected to adjacent regions causing a decrease of precipitation. Another explanation could be that the cooler surface temperatures through irrigation lead to less convective processes above the irrigated areas. In the end, precipitation-irrigation effects are more complex than eg. irrigation-temperature effect and require a different experiment setting as we pointed out also to the comment of Jozsef Szilagyi. We added these hypotheses in our text. | line 452: In IT, precipitation (Fig. 16c) occurs on the 2nd and 3rd of August at very low rates, which can be neglected, and on the 5th, 6th, and 7th of August at higher rates up to 4.5 mm day-1 in the non-irrigated simulation and 2.5 mm day-1 in the irrigated simulation. However, these precipitation rates are very low and affect the soil moisture with a small increase from 0.45 of wsmx to 0.47 of wsmx in the non-irrigated simulation on the 5th of August. **As in subsubsection 4.4.2, the effect of irrigation on precipitation is unclear during the heat wave. In Fig. 16c precipitation increases on the 5th, decreases on the 6th, and stays the same on the 7th of August. A possible explanation for the precipitation increase might be the higher evapotranspiration rate and higher relative humidity (as shown in subsubsection 4.4.2). However, the temperature changes through irrigation can also affect wind patterns, so that the humidity is advected outside our analysis region IT. Further, the cooling effect of irrigation on the surface temperature and near-surface temperature leads to less convective processes, which might have developed in the not-irrigated simulation on the 6th of August.** |
| Figure A1. | Referee 1 | Figure A1. Too small, please, make it horizontal | Thank you for your suggestion. We applied it. | Rotated and increased Figure A1 |
| 575 | CE Juan A. Añel | Also, although, in general, you have done good work with the repositories for data, you fail to do it with the data from http://www.scia.isprambiente.it/. This webpage is not a long-term, trustable repository. Moreover, you provide little information about how to get the exact data you have used, as you only include a generic link to the home page. Therefore, we have to ask you to deposit the data that you have used in one of the acceptable repositories listed in our policy. | Thank you for pointing out this weakness in our study. We improved the description of the download of the observational data from the SCIA website and added the used observation data to our repository along with a copyright statement on zenodo after receiving the permission of ISPRA (who hosts the SCIA data). | line 601: The scripts used to produce the results presented in this paper are archived on Zenodo ( https://doi.org/10.5281/zenodo.7889384 ), as is the simulation data together with the observational data from SCIA (https://doi.org/10.5281/zenodo.7867328) after we received their permission. Originally, we downloaded the observation data from http://www.scia.isprambiente.it/ using the API http://193.206.192.214/servertsutm/serietemporali400.php accessed on 14/10/2022. The code can be cited as Asmus and Buntemeyer (2023), the data of this study can be cited as Asmus (2023). |
| 572 | CE Juan A. Añel | You do not provide a repository for the code of REMO2020-iMOVE, and you state that for it, an email address should be contacted for it. We can not accept this. You must publish the code of the new developments to the model and all the model code in one of the suitable repositories. Despite the request by the handling topical editor regarding this issue, and according to our records in the system, you did not address it. | We provided the REMO2020-iMOVE code with the initial submission on request of the handling topic editor. Therefore, the code was accessible to the reviewers and fulfilled the GMD Code & Data policy for restricted codes. The code is restricted due to reason beyond us authors, therefore, we cannot publish it. This issue was solved in a private discussion on 28/08/2023. | - |

**Further corrections by the authors**

| line | comment from | comments from referee/community/executive editor | author's response | author's changes in manuscript (in bold) with line number in revised version |
|---|---|---|---|---|
| Figure 1 caption | Authors | Caption of Fig.1 is not clear enough. | We improved the caption of Fig. 1 and made clear that the two bars show one example grid cell without the irrigation parameterization and one with the irrigation parameterization. | Figure1. Fractions of one example model grid cell in REMO2020-iMOVE + FLake with and without irrigation |
| 309 | Authors | In Bjorneberg, we found the number of irrigation hours specifically for channel irrigation with 12- 24 h, and not as we stated as 6 - 12 h. | We corrected this mistake and verfied the numbers with another source. | line 231: Following Bjorneberg (2013) **and Zucaro (2014) channel irrigation is performed for up to 24 hours depending on the channel width and length,** we chose 10 h irrigation time for our experiment. |
| 122 | Authors | W found a mistake in the definition of fast drainage in REMO2020-iMOVE. Fast drainage occurs from 90 %. | We corrected it in the text. | line 122: Drainage occurs for soil moisture larger than 5 % of wsmx. Between 5 % and **90** % of wsmx, drainage is slow. If the soil moisture is larger than 90 % of wsmx, the drainage is fast (Kotlarski, 2007). |
| 511 | Authors | The dissertation on a 5-layer scheme for REMO was recently published. | Therefore, we adjusted the information about it in the text and used it as refer | line 528: For the future, for irrigation studies, we recommend the representation of the soil hydrology with a multiple layer scheme as for WRF (Valmassoi et al., 2019) or CLM (Lawrence et al., 2019; Ozdogan et al., 2010) it already exists, and which **was currently developed for REMO2015 (Abel, 2023)**. |
| 512 | Authors | Rai et al. did not use the 5-layer scheme, they used REMO2015-iMOVE. | We added Rai et. al 2022 to a better fitting place. | line 549: The missing evaporative cooling of the transpiration of vegetation leads to increasing 2 m temperatures. This effect was already observed by Wilhelm et al. (2014) **and Rai et al. (2022).** |
| 106, 617 | Authors | Hoffmann et al. was published recently published. | We added the correct reference of the published paper version (before it was refered to the preprint). | line 106: For this experiment, the definition and distribution of PFTs are based on the land cover maps of the European Space Agency Climate Change Initiative (ESA-CCI) (Reinhart et al., 2022; Hoffmann et al., **2023**). line 654: Hoffmann, P., Reinhart, V., Rechid, D., de Noblet-Ducoudré, N., Davin, E. L., Asmus, C., Bechtel, B., Böhner, J., Katragkou, E., and Luyssaert, S.: High-resolution land use and land cover dataset for regional climate modelling: historical and future changes in Europe,655 Earth System Science Data, 15, 3819–3852, **https://doi.org/10.5194/essd-15-3819-2023, 2023.** |
| - | Authors | Unclearity in doi in references. | We improved the doi citations in the references. | eg. Yao, Y., Vanderkelen, I., Lombardozzi, D., Swenson, S., Lawrence, D., Jägermeyr, J., Grant, L., and Thiery, W.: Implementation and Evaluation of Irrigation Techniques in the Community Land Model, Journal of Advances in Modeling Earth Systems, 14, e2022MS003 074, **https://doi.org/10.1029/2022MS003074**, 2022. |